

# Gluon condensates and effective gluon mass

Jan Horak[1], Friederike Ihssen[1], Joannis Papavassiliou[2],
Jan M. Pawlowski[1,3], Axel Weber[1,3,4] and Christof Wetterich[1]

**1** Institut für Theoretische Physik, Universität Heidelberg,
Philosophenweg 16, 69120 Heidelberg, Germany
**2** Department of Theoretical Physics and IFIC, University of Valencia and CSIC,
E-46100, Valencia, Spain
**3** ExtreMe Matter Institute EMMI, GSI, Planckstr. 1, 64291 Darmstadt, Germany
**4** Instituto de Física y Matemáticas, Universidad Michoacana de San Nicolás de Hidalgo,
Edificio C-3, Ciudad Universitaria, A. Postal 2-82, 58040 Morelia, Michoacán, Mexico

## Abstract

Lattice simulations along with studies in continuum QCD indicate that non-perturbative quantum fluctuations lead to an infrared regularisation of the gluon propagator in covariant gauges in the form of an effective mass-like behaviour. In the present work we propose an analytic understanding of this phenomenon in terms of gluon condensation through a dynamical version of the Higgs mechanism, leading to the emergence of color condensates. Within the functional renormalisation group approach we compute the effective potential of covariantly constant field strengths, whose non-trivial minimum is related to the color condensates. In the physical case of an $SU(3)$ gauge group this is an octet condensate. The value of the gluon mass obtained through this procedure compares very well to lattice results and the mass gap arising from alternative dynamical scenarios.



# 1  Introduction

Yang-Mills theory exhibits a mass gap, in spite of the fact that the fundamental degrees of freedom are massless at the level of the classical action. While perturbation theory is based on massless gluons, non-perturbative quantum fluctuations lead to exponentially decaying correlation functions for gauge invariant observables, which are characteristic of massive excitations. The lightest excitations are glueballs [1, 2], and the lightest glueball mass sets the mass gap or confinement scale. This dynamical emergence of a mass gap in the gauge sector of QCD has been established by numerous lattice studies, see e.g. [3–7], and continuum studies, see e.g. [8–16].

In a gauge fixed version of QCD the effects of the mass gap manifest themselves through the appearance of distinctive patterns in the infrared momentum region of correlation functions. Most of the related investigations have been performed in Landau gauge QCD. In particular the infrared behaviour of the gluon propagator in Landau gauge has been explored within large-volume lattice simulations [17–26] and non-perturbative functional methods, such as Dyson-Schwinger equations (DSEs) [27–31] and the functional renormalisation group (fRG) [32–35]. In combination, these investigations have led to a coherent picture: with exception of the deep infrared regime far below the confinement scale $\Lambda_{QCD}$, the results obtained for the gluon propagator in the non-perturbative domain are in excellent agreement. In particular, they are found to be well compatible with a description in terms of an effective gluon mass. Put differently, they show the dynamical emergence of a mass gap in the gluon propagator, and in higher order correlation functions.

The precise relation between the gluon mass in gauge fixed QCD and the physical mass gap in Yang-Mills theory still eludes us. Nonetheless, in covariant gauges a mass gap in the gluon propagator is required for quark confinement to occur, as has been established through the study of the Polyakov loop expectation value in [36, 37].

This situation asks for the identification and investigation of potential mechanisms which

are able to create an effective gluon mass term. Commonly, gauge boson masses are generated by the formation of condensates, even in the absence of fundamental scalar fields. The textbook implementation of such a scenario is realised within the theory of superconductivity. There, the massive photon associated with the Meissner effect is linked to the condensation of the Cooper pairs, see e.g. [38, 39], and references therein. In pure Yang-Mills theory, a potential connection between the effective gluon mass and gluon condensates of dimension four has mostly been discussed within the operator product expansion (OPE) [40–42]. It has been argued in [43] that a non-perturbative condensate of composite color octets in QCD leads to a simple description of gluon masses by the Higgs mechanism. In this scenario, the massive gluons can be identified with the lowest mass vector mesons, with a rather successful phenomenology [44, 45].

The present work is a first fRG study of a potential dynamical emergence of the effective mass in the gauge fixed gluon propagator in QCD color condensates. This condensate is computed from the Euclidean effective potential of a constant field strength $F_{\mu\nu}$ as in [46], with precision ghost and gluon propagators obtained within the fRG [47]. We find minima and saddle points for finite non-zero $F_{\mu\nu}$. The minimum value of $F_{\mu\nu}$ is related to an effective gluon mass, and the final color blind result is obtained from an average over color directions. Our computation of the effective gluon mass agrees very well with lattice results and results obtained from alternative dynamical scenarios within the error bars, despite the qualitative nature of the computation. The present study serves as a promising starting point for a systematic exploration of the connection between gluon condensates and gluon mass gap.

## 2  Gluon Condensates

Gluon condensation can be described by non-vanishing expectation values of composite operators, such as the field strength squared, $F_{\mu\nu}F_{\mu\nu}$, being a scalar under Lorentz transformations. In terms of the free energy or effective action of QCD, this entails that quantum effects would trigger a non-trivial potential in these condensates, with the possibility of capturing also the dynamics of the respective interaction channel. In this context, the classical action of Yang-Mills theory is the first (trivial) term of such a non-trivial potential,

$$S_A[A] = \frac{1}{2}\int \operatorname{tr} F_{\mu\nu}^2 = \frac{1}{4}\int F_{\mu\nu}^a F_{\mu\nu}^a, \tag{1}$$

with the field strength $F_{\mu\nu} = F_{\mu\nu}^a t^a$, where

$$F_{\mu\nu}^a = \partial_\mu A_\nu^a - \partial_\nu A_\mu^a + g_s f^{abc} A_\mu^b A_\nu^c. \tag{2}$$

In eq. (1), the trace is taken over the fundamental representation, with $\operatorname{tr}(t^a t^b) = \frac{1}{2}\delta^{ab}$, and $a, b = 1, ..., N_c^2 - 1$ for the gauge group $SU(N_c)$. Since ensuing computations involve covariantly constant field strengths, for which $[D, F] = 0$, we also report the standard relation

$$F_{\mu\nu} = \frac{i}{g_s}[D_\mu, D_\nu], \quad \text{with} \quad D_\mu = \partial_\mu - ig_s A_\mu, \tag{3}$$

with the algebra valued field $A_\mu = A_\mu^a t^a$.

### 2.1  Color condensates

Color condensates [43, 48–50] could render the gluons massive through a dynamical realisation of the Higgs mechanism. Note that, strictly speaking, a local gauge symmetry cannot be

broken spontaneously. Nonetheless, as is well-known from the description of the electroweak sector of the Standard Model, the language of spontaneous symmetry breaking in a fixed gauge can be particularly useful, and will be employed in what follows.

Below we discuss a color condensate operator, derived from $F_{\mu\nu}$ in the case of the physical gauge group $SU(3)$. Generally, a possible condensate operator of dimension four is given by the traceless hermitian $N_c \times N_c$ matrices

$$\chi^{AB} = \left( F_{\mu\nu}^{AC} F_{\mu\nu}^{CB} - \frac{1}{N_c} F_{\mu\nu}^{CD} F_{\mu\nu}^{DC} \delta^{AB} \right), \tag{4}$$

where $A, B, C, D = 1, ..., N_c$ are color indices in the fundamental representation, $F_{\mu\nu}^{AB} = F_{\mu\nu}^a (t^a)^{AB}$. The subtraction of the diagonal term makes the operator traceless, $\chi^{AA} = 0$, and for $N_c = 3$ this is an octet operator. In terms of the field strength components $F_{\mu\nu}^a$, the condensate in eq. (4) reads,

$$\chi^{AB} = \frac{1}{2} F_{\mu\nu}^a F_{\mu\nu}^b \left( \{t^a, t^b\}^{AB} - \frac{1}{N_c} \delta^{ab} \delta^{AB} \right). \tag{5}$$

We note in passing that the above operator is only present for $N_c \geq 3$. It vanishes in $SU(2)$, as the symmetric group invariant vanishes, $d^{abc} = \operatorname{tr} t^a \{t^b, t^c\} = 0$. This already suggests that in a realistic condensation scenario leading to a gluon mass gap, eq. (4) should be augmented with further color condensate operators.

Introducing the composite color condensate field $\chi^{AB}$, the quantum effective action $\Gamma$ will contain an induced kinetic term,

$$\Gamma_\chi = Z_\chi \int_x (D_\mu \chi)^{AB} (D_\mu \chi)^{BA}, \tag{6}$$

with a wave function renormalisation $Z_\chi$. For a non-zero expectation value $\langle \chi^{AB} \rangle$, this induces a mass term for some of the gluons,

$$m_A^2 \propto Z_\chi g_s^2 \langle \chi \rangle^2. \tag{7}$$

Mass terms for all gluons in $SU(3)$ require condensates of more than one octet in different directions since at least a $U(1) \times U(1)$-subgroup remains unbroken, as for example in [43, 48–50]. This argument also applies to higher gauge groups, $N_c \geq 3$, and we have already pointed out in this context that the color condensate operator eq. (5) vanishes for $N_c = 2$. Besides different mass terms, octet condensates can also induce different effective gauge couplings for different gluons, due to terms in the effective action, see e.g. [51, 52],

$$\int_x F_{\mu\nu}^{AB} \chi^{BC} F_{\mu\nu}^{CA}. \tag{8}$$

This closes our discussion of color condensation in Yang-Mills theories.

## 2.2 Color condensates and the field strength tensor

The flow equation approach with dynamical composite fields such as the color condensate field discussed in the last section is well understood. It has been introduced and discussed in [53–62], for applications to QCD see [59, 63–67] and the review [35]. However, full computations including the composite field $\chi^{AB}$ require a substantial effort, and will be considered elsewhere.

In the present work we restrict ourselves to a qualitative study, whose principal aim is to gather insights on the possible rôle of non-singlet condensates in the confining dynamics. This is done by building on results for the condensation of the field strength tensor within functional

renormalisation group investigations in [46, 68, 69]. Such a colored expectation value of $F_{\mu\nu}$ is linked to non-vanishing expectation values of the color condensate operator $\chi$ in eq. (4) as well as potential non-vanishing expectation values of further color condensate operators. Hence, $\langle F_{\mu\nu} \rangle$ can be used to describe the dynamical emergence of the effective gluon mass via color condensates, for details see Section 2.3.

We emphasise that a description in terms of $F_{\mu\nu}$ and its expectation value makes it difficult to include the full dynamics of the color condensate sector as well as the condensation pattern, as this requires the computation of the dynamics of higher order terms in $F_{\mu\nu}$ and covariant derivatives. We also note that such an expansion about $\langle F_{\mu\nu} \rangle$ works naturally for observables or more generally, expectation values of gauge invariant operators. There, singling out a color direction is simply a means of computation. In turn, for gauge-variant expressions the expansion about a non-trivial configuration mixes with the gauge fixing, and it is difficult to undo the color selection quantitatively. Still, it can be done with an additional color averaging $\langle \cdot \rangle_{\text{av}}$, which can be implemented systematically. As this concerns the understanding and underlying structure of our work, we further explain this with two simple examples. While important, it is not in our main line of reasoning and hence is deferred to Appendix A.

Note, that such an averaging is to date always implied in lattice simulations of gauge fixed correlation functions as well as in most computations in functional QCD using an expansion about the only color-symmetric background, $\langle F_{\mu\nu} \rangle = 0$. The intricacies mentioned above only occur for a quantitative implementation in an expansion about a colored background. It is the current lack of a quantitatively reliable averaging procedure, that causes the current investigation to be of qualitative nature, and constitutes our largest source of systematic error.

In the present work, we compute the respective gauge invariant effective potential $\mathcal{W}_{\text{eff}}(F_{\mu\nu})$ for constant field strength $F_{\mu\nu}$ from the effective action $\Gamma[A]$,

$$\mathcal{W}_{\text{eff}}(F_{\mu\nu}) = \frac{1}{\mathcal{V}} \Gamma_k[A(F_{\mu\nu})], \tag{9}$$

with the space time volume $\mathcal{V}$.

Specifically, we choose gauge fields with the following constant self-dual field strengths: the components $F_{\mu\nu} = 0$ for $\mu\nu \neq 01, 10, 23, 32$ vanish, and we have

$$F_{01} = F_{23} = \frac{F^a}{2g_s} t^a, \quad F_{01}^a = \frac{F^a}{2g_s}, \quad F^a = F n^a, \tag{10a}$$

with a constant vector $n^a$ with $n^a n^a = 1$. The field strength eq. (10a) can be generated from the gauge fields

$$A_\mu^a = -\frac{1}{2} F_{\mu\nu}^a x_\nu. \tag{10b}$$

Evidently, the configuration is self-dual,

$$F_{\mu\nu} = \tilde{F}_{\mu\nu}, \quad \text{with} \quad \tilde{F}_{\mu\nu} = \frac{1}{2} \epsilon_{\mu\nu\rho\sigma} F_{\rho\sigma}, \tag{10c}$$

and is covariantly constant, $[D_\rho, F_{\mu\nu}] = 0$.

The classical action and the classical potential $\mathcal{W}_{\text{cl}}$ as well as the color condensate eq. (4) is obtained from the field strength squared, which reads for the configuration eq. (10),

$$F_{\mu\nu} F_{\mu\nu} = \frac{F^2}{g_s^2} (n^a t^a)^2, \quad F_{\mu\nu}^a F_{\mu\nu}^a = \frac{1}{g_s^2} F^2. \tag{11}$$

For example, for the configuration eq. (10) with eq. (11), the classical potential reduces to

$$\mathcal{W}_{\text{cl}}(F^a) = \frac{1}{2} \text{tr} \, F^a F^b \, t^a t^b = \frac{1}{4g_s^2} F^2, \tag{12}$$

where tr is the group trace in the fundamental representation as in eq. (1). From now on we only consider configurations of the type eq. (10), and hence $\mathcal{W}_{\text{eff}}$ will be written as a function of $Fn^a$, that is $\mathcal{W}_{\text{eff}}(F^a)$ instead of $\mathcal{W}_{\text{eff}}(F_{\mu\nu})$. The factor $1/g_s^2$ in eq. (11) reflects the RG-scaling of the field strength, and has been introduced for convenience. Moreover, as both the gauge fields and the field strength in eq. (10b) point in direction $n^a$ of the algebra, they can be rotated into the Cartan subalgebra without loss of generality.

Below, we briefly discuss $SU(2)$ and $SU(3)$ gauge groups, the former case as the simplest example, the latter case for its physical relevance:

In the $SU(2)$ gauge group, the Cartan subalgebra is generated by $t^3 = \sigma^3/2$ and the self-dual field strength eq. (10) is given by

$$F_{01} = F_{23} = \frac{F}{2g_s}\, t^3\,. \tag{13}$$

We have already discussed above that in $SU(2)$ the symmetric group invariant $d^{abc}$ vanishes, and hence $\chi_{SU(2)}^{AB} = 0$, implying $(F_{\mu\nu}F_{\mu\nu})^{AB} = F_{\mu\nu}^a F_{\mu\nu}^a \delta^{AB}/4$ for all configurations. For eq. (13) we find

$$(F_{\mu\nu}F_{\mu\nu})^{AB} = \frac{F^2}{4g_s^2}\delta^{AB}\,. \tag{14}$$

The explicit computation in this work is done for the physical gauge group $SU(3)$ with the Cartan generators $t^3, t^8$. These are related to the Gell-Mann matrices by $t^a = \lambda^a/2$, the respective vector $n$ has the components $n^a = 0$ for $a \neq 3, 8$. A self-dual field strength eq. (10) is given by

$$F_{01} = F_{23} = \frac{F}{2g_s}\bigl(n^3 t^3 + n^8 t^8\bigr)\,. \tag{15}$$

The octet condensate operator eq. (4) for the configuration eq. (15) reads

$$
\begin{aligned}
\chi^{AB} &= \frac{F^2}{2g_s^2}\Bigl[n^a n^b\,\{t^a, t^b\}^{AB} - \frac{1}{3}\delta^{AB}\Bigr]\\
&= \frac{F^2}{2g_s^2}\delta^{AB}\bigl[\delta^{A1}\nu_+ + \delta^{A2}\nu_- + \delta^{A3}\nu_3\bigr]\,,
\end{aligned} \tag{16}
$$

where

$$\nu_\pm = \frac{1}{2}\left(\frac{n^8}{\sqrt{3}} \pm n^3\right)^2 - \frac{1}{3}\,, \qquad \nu_3 = \frac{2}{3}(n^8)^2 - \frac{1}{3}\,, \tag{17}$$

where the trace(less) condition, $\chi^{AA} = 0$, translates into $\nu_+ + \nu_- + \nu_3 = 0$ with $(n^3)^2 + (n^8)^2 = 1$.

Non-vanishing octet condensate expectation values are in one to one correspondence to non-trivial expectation values of its corresponding gauge-invariant eigenvalues. Hence, a non-trivial expectation value of the field strength triggers one for the octet condensate $\chi^{AB}$ and other color condensate operators. Therefore, in Section 4, we compute the effective potential for covariantly constant field strength or rather $\mathcal{W}_{\text{eff}}[Fn^a]$ for the field strength amplitude $Fn^a$ defined in eq. (10a), and the constant algebra element $n^a t^a$ is rotated into the Cartan subalgebra leading to eq. (15). The respective effective potential is shown in Figure 1 for the physical $SU(3)$ case with the two Cartan components $F_{01}n^3$ and $F_{01}n^8$.

Our explicit computation of the effective gluon mass is based on an expansion about the minimum $\langle F \rangle(n^a)$ in the three-direction with $n^a = \delta^{a3}$. In $SU(2)$ this is the Cartan direction,

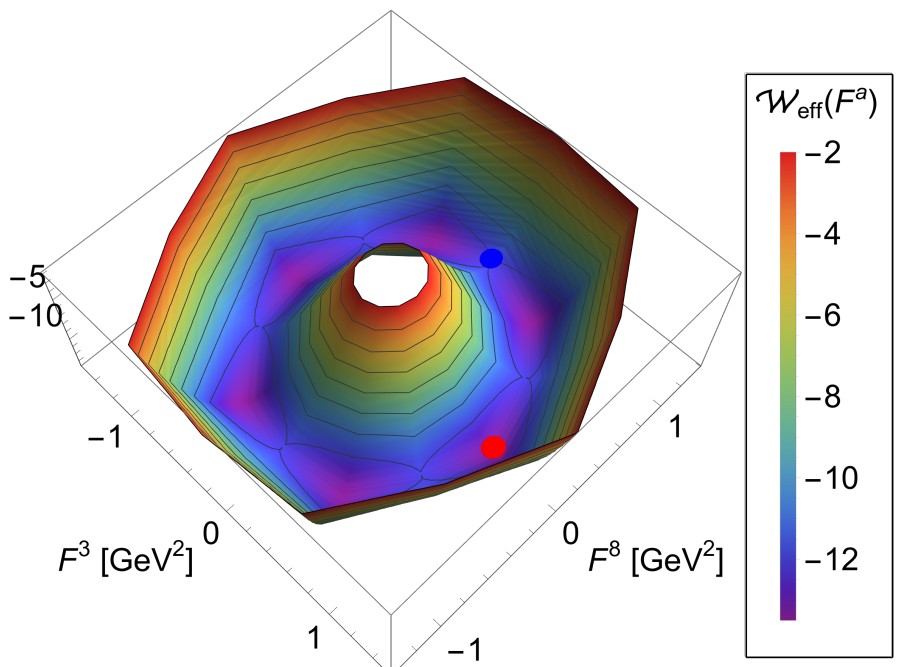

Figure 1: Effective potential $\mathcal{W}_{\text{eff}}(F^a)$ in the plane spanned by the Cartan subalge-bras. The position of the non-trivial global minimum is highlighted in red.

and in $SU(3)$ one of the absolute minima points in the three-direction, see Figure 1. Then, the expansion about the minimum reads

$$F^a_{01} = F^a_{23} = \frac{\langle F \rangle}{2g_s} \delta^{a3} + \mathcal{O}(a),$$ (18)

for both gauge groups, where $a_\mu$ is the gauge field, that carries the fluctuations about the field strength expectation value. With eq. (10b) we can deduce a gauge field, that generates eq. (18). We find,

$$A^a_\mu = \frac{\langle F \rangle}{4g_s}\Big(x_0\delta_{\mu 1} - x_1\delta_{\mu 0} + x_2\delta_{\mu 3} - x_3\delta_{\mu 2}\Big) + a^a_\mu,$$ (19)

which points in the same Cartan direction as the field strength. The fluctuation field $a_\mu$ carries the dynamics of the gauge field, leading to the $\mathcal{O}(a)$-terms in eq. (18). Within this setting we shall derive our estimates for the effective gluon mass as well as discuss constraints and bounds for this mass.

## 2.3 Color condensates and the gluon mass gap

It is left to discuss the emergence of an effective gluon mass term in the presence of gluon condensates via the expectation value $\langle F^a_{\mu\nu} \rangle \propto \delta^{a3}$ in Equation (18), or any other algebra direction. This expectation value is computed from the effective potential $\mathcal{W}(F^a)$ introduced in section 2.2.

Expanding the effective potential in powers of the fluctuation field gauge field $a_\mu$ leads to contributions to the $n$-point functions of the gauge field, including the two-point function. However, neither a contribution to the mass operator $a^a_\mu a^a_\mu$ is generated, nor do we obtain mass terms in specific algebra directions. In particular, no mass contribution in the Cartan $a = 3$ direction is induced, as is readily shown for $SU(2)$.

While the effective potential does not contribute to the effective mass term, the latter receives contributions from other terms in the full, gauge invariant quantum effective action $\Gamma[A]$. Such an action can be defined within the background field approach, which will be detailed in Section 3. For the time being we simply assume its existence and consider the higher order term

$$\Gamma_F[A] = \frac{Z_F}{4} \int_x (D_\mu F_{\nu\rho})^a (D_\mu F_{\nu\rho})^a \,, \tag{20}$$

where $Z_F$ is the wave function renormalisation of the condensate term. Equation (20) is the lowest order term that generates an effective gluon mass term within an expansion about the condensate $\langle F \rangle$. An obvious generalisation of eq. (20) is provided by

$$\frac{1}{4} \int_x (D_\mu F_{\rho\sigma})^a Z_F{}^{ab}_{\rho\sigma\alpha\beta}(F_{\mu\nu})(D_\mu F_{\alpha\beta})^b \,, \tag{21}$$

with $Z_F{}^{ab}_{\rho\sigma\alpha\beta}(0) = Z_F \delta_{\rho\alpha}\delta_{\sigma\beta}\delta^{ab}$. In the following we will use the approximation

$$Z_F{}^{ab}_{\rho\sigma\alpha\beta}(\langle F_{\mu\nu}\rangle) \approx Z_F{}^{ab}_{\rho\sigma\alpha\beta}(0), \tag{22}$$

hence only considering the term eq. (20).

Equation (20) leads to an effective gluon mass, but does not contribute to (covariantly constant) solutions of the equations of motions as its first field derivative vanishes for covariantly constant field strengths. The relevant contribution to the effective gluon mass term is obtained by expanding eq. (20) in powers of the gauge field, while treating the field strength within the expansion eq. (18). To that end we conveniently recast eq. (20) into the form

$$\Gamma_F[A] = -\frac{Z_F}{2} \int_x F^{CB}_{\nu\rho}(D^2)^{BA} F^{AC}_{\nu\rho} \,, \tag{23}$$

where the factor $1/2$ in eq. (20) is now carried by the trace in the fundamental representation. The $\mathcal{O}(A^2)$ term is given by

$$\Gamma_F[A] = \frac{Z_F}{2} g_s^2 \int_x (F_{\nu\rho}F_{\nu\rho})^{AB} (A_\mu A_\mu)^{BA} + \cdots \,, \tag{24}$$

and we expand $(F_{\nu\rho}F_{\nu\rho})^{AB}$ about the field strength expectation value eq. (18). This implies a non-vanishing condensate expectation value for eq. (4) as well as non-vanishing values for other color condensate operators. The expansion about eq. (18) leads us to

$$(F_{\nu\rho}F_{\nu\rho})^{AB} = \frac{1}{g_s^2} \langle F \rangle^2 [(n^a t^a)^2]^{AB} + \mathcal{O}(a). \tag{25}$$

We drop the higher order terms in eq. (25) and insert it in eq. (24), to wit,

$$\Gamma_F[A] \simeq \frac{Z_F}{2} \langle F \rangle^2 \int_x \mathrm{tr}(t^3)^2 A_\mu^2 + \cdots \,, \tag{26}$$

with the group trace in the fundamental representation. Now we evaluate eq. (26) for the configurations eq. (10), which leads to our final expression for the effective gluon mass triggered by an expectation value of the field strength proportional to $t^3$. For general gauge groups, eq. (26) is not color blind, which originates in the colored expansion point. It can be used to deduce the color blind mass by a color average discussed in Appendix A.

Before we come to our final color blind estimates, we exemplify eq. (26) within $SU(2)$ and $SU(3)$. We first discuss the simple example of an $SU(2)$ gauge group. There, the configuration eq. (18) leads to an $F_{\mu\nu}^2$ that is proportional to the identity tensor $\mathbb{1}$ in the algebra, as $4(t^3)^2 = \mathbb{1}$. Indeed, as discussed below eq. (13), general field strength tensors lead to diagonal $F_{\mu\nu}^2$. In summary, in $SU(2)$, a field strength condensate in the $t^3$-direction leads to

$$\Gamma_F[A] \simeq \frac{1}{2}m_3^2 \int_x A_\mu^a A_\mu^a + \cdots, \quad m_3^2 = \frac{Z_F}{8}\langle F\rangle^2, \tag{27}$$

with a uniform mass $m_3$ for all gluons. The subscript indicates that, while uniform, the mass is generated by $\langle F_{\mu\nu}^a\rangle \propto \delta^{a3}$. Importantly, eq. (27) entails that a color condensate leads to gluons with an effective mass. However, the current procedure with an expansion about a non-vanishing field strength does not allow to directly infer the full effective gluon mass obtained in a color blind computation from $m_3^2$ in eq. (27). At this stage we only can offer estimates, whose derivation is deferred to the end of the present section.

Before we come to these estimates, we proceed with the $SU(3)$ example. There, we also use the Cartan-valued configuration eq. (18) (with $n^8 = 0$) as one of the absolute minima in the full effective potential $\mathcal{W}_{\text{eff}}$ points in this direction, see Figure 1. In contradistinction to $SU(2)$, the square $4(t^3)^2$ is not the identity matrix in the algebra, but a projection onto the first two colors,

$$[(t^3)^2]^{AB} = \frac{1}{4}\delta^{AB}\left(\delta^{A1} + \delta^{A2}\right). \tag{28}$$

As expected, the expansion about a minimum of the field strength, related to one about the octet condensate eq. (4), breaks color, and indeed, the gluon with the third color is massless if only considering contributions from $\Gamma_F$. Hence, while the present expansion shows, that the gluons acquire an effective mass term $\propto \delta^{ab}$, the relation of its *necessarily color blind* value $m_A^2$ to the color-sensitive masses derived here is not straightforward.

Therefore, in the present work we simply deduce self-consistency constraints for the effective mass $m_A^2$ starting with the gluon mass $m_3^2$, inferred from a field strength in the $t^3$ direction. To begin with, color symmetry can be restored by averaging over global color rotations as always implied in lattice simulations as well as in most computations in functional QCD. After this averaging, all masses are identical and non-vanishing. A color average of eq. (26) leads us to

$$\Gamma_{A^2}[A] = \frac{Z_F}{2}f_{\text{av}}(N_c)\langle F\rangle^2 \int_x A_\mu^a A_\mu^a, \tag{29}$$

with $f_{\text{av}}(N_c)$ encodes the color average of the factor $(t^3)^2$ in eq. (26),

$$f_{\text{av}}(N_c) = \langle(t^3)^2\rangle_{\text{av}}. \tag{30}$$

The color average in eq. (30) necessarily leads to a color insensitive sum over all generators squared in the fundamental representation, which is simply the second Casimir $C_2(N_c) = (N_c^2 - 1)/(2N_c)$ times the identity matrix. Moreover, there is an undetermined prefactor $c_{\text{av}}(N_c)$, which leads us to

$$\left\langle(t^3)^2\right\rangle_{\text{av}} = c_{\text{av}}(N_c)\sum_{a=1}^{N_c^2-1}(t^a)^2 = c_{\text{av}}(N_c)C_2(N_c)\mathbb{1}. \tag{31}$$

In the present work we will only provide constraints for $c_{\text{av}}(N_c)$ and hence for $f_{\text{av}}(N_c)$ in eq. (30). For example, a 'natural' bound for the averaging factor is unity, $c_{\text{av}}(N_c) \leq 1$.

In summary we arrive at

$$m_A^2 = \frac{Z_F}{2} f_{\text{av}}(N_c) \langle F \rangle^2 \,. \tag{32}$$

In Section 3.3 we will show, that self-consistency of the averaging in the large $N_c$ limit entails that in this limit $f_{\text{av}}(N_c) \propto N_c$. Indeed, this limit holds true for $N_c$-independent $c_{\text{av}}$. In particular this includes the case, where we saturate the 'natural' bound $c_{\text{av}} = 1$, leading to

$$f_{\text{av}} = (N_c^2 - 1)/(2 N_c) \,. \tag{33}$$

For this saturation $f_{\text{av}}$ we obtain

$$m_A^2 = \frac{Z_F}{4} \frac{N_c^2 - 1}{N_c} \langle F \rangle^2 \,. \tag{34}$$

Equation (34) will eventually yield our estimate of the effective gluon mass. In Section 3, we present the formalism employed for working with the constant field strength configurations in eq. (11). The computation of the minimum position $F^a = \langle F \rangle \, n^a$ is detailed in Section 4, and an estimate of the wave function of the condensate together with the result for the mass gap is presented in Section 5.

# 3  Background field approach

The condensate $\langle F \rangle$ for the field strength configuration of eq. (15) is given by the minimum of an effective potential $\mathcal{W}_{\text{eff}}(F \, n^a)$, derived from a gauge invariant effective action $\Gamma[A]$, see eq. (9). Such an action is defined in the background field approach [70], building on a linear decomposition of the full gauge field $A_\mu$ into a fluctuating and background field. This linear split is given by $A_\mu = \bar{A}_\mu + a_\mu$, where $a_\mu$ denotes the fluctuation field and $\bar{A}_\mu$ the background field. On the quantum level, this relation has to be augmented with the respective wave function renormalisations $Z_{\bar{A}} = Z_{g_s}^{-2}$ for the background field $\bar{A}_\mu$ and $Z_a$ for the fluctuation field $a_\mu$, as the two fields carry different RG scalings: As indicated above, the background field scales inversely to the strong coupling, while the fluctuation field carries the RG-scaling of the gauge field in the underlying gauge without background field. The gauge fixing condition involves the background field,

$$\bar{D}_\mu a_\mu = 0 \,, \tag{35}$$

with the background covariant derivative $\bar{D} = D(\bar{A})$, see eq. (3). Note, that eq. (35) is invariant under background gauge transformations,

$$a \to a + i[\omega, a] \,, \qquad \bar{A} \to \bar{A} + \frac{1}{g_s} \bar{D} \omega \,, \tag{36}$$

implying a standard gauge transformation for the full gauge field: $A_\mu \to A_\mu + (1/g_s) D \omega$. Consequently, the full gauge-fixed classical action is invariant under eq. (36), and so is the full effective action $\Gamma[\bar{A}, a]$. Moreover, the single-field background field effective action $\Gamma[A] := \Gamma[A, 0]$ is gauge invariant and can be expanded in gauge invariant operators. For this reason, it also allows for a more direct access to observables. In what follows we use the potential condensate background eq. (10).

### 3.1 Background field effective action

The gauge invariance of the background field effective action allows us to embed the momentum-dependent kinetic terms and vertices in an expansion about a vanishing gauge field in full gauge invariant terms that reduce to the original ones for $A_\mu \to 0$. An important example is given by the (transverse) kinetic term of the gauge field, see e.g. [36,37,46],

$$\Gamma[A] \propto \frac{1}{2} \int_p A_\mu^a(p) Z_A(p^2) p^2 \Pi_{\mu\nu}^\perp(p) A_\nu^a(-p) \,, \tag{37}$$

with the abbreviation $\int_p = \int d^4p/(2\pi)^4$, and the transverse and longitudinal projection operators

$$\Pi_{\mu\nu}^\perp(p) = \delta_{\mu\nu} - \frac{p_\mu p_\nu}{p^2} \,, \qquad \Pi_{\mu\nu}^\parallel(p) = \frac{p_\mu p_\nu}{p^2} \,. \tag{38}$$

The kinetic operator $Z_A(p^2)p^2$ is identified as the $A_\mu \to 0$ limit of the second field derivative of a gauge invariant term in the effective action $\Gamma[A]$. This leads us straightforwardly to the parametrisation

$$\Gamma[A] = \frac{1}{2} \int \mathrm{tr} F_{\mu\nu} f_{A,\mu\nu\rho\sigma}(D) F_{\rho\sigma} + \cdots \,, \tag{39a}$$

with the split

$$f_{A,\mu\nu\rho\sigma}(D) = \frac{1}{2} Z_A(\Delta_s)(\delta_{\mu\rho}\delta_{\nu\sigma} - \delta_{\mu\sigma}\delta_{\nu\rho}) + F_{\gamma\delta} f_{A,\gamma\delta\mu\nu\rho\sigma}(D) \,. \tag{39b}$$

In eq. (39b), we have introduced the spin-$s$ Laplacians

$$\Delta_0 = -D^2 \,, \qquad \Delta_{1,\mu\nu} = \mathcal{D}_{T,\mu\nu} = -D^2 \delta_{\mu\nu} + 2i g_s F_{\mu\nu} \,, \tag{39c}$$

see also eq. (86). Equation (39b) represents the most general parametrisation for a covariant function coupled to two field strengths. Since $f_{A,\gamma\delta\mu\nu\rho\sigma}$ is a function of the covariant derivative $D$, higher order terms in the field strength tensor are contained in the second term of eq. (39b). For $A_\mu = 0$, all these decompositions reduce to their the momentum-dependent versions. In particular, the kinetic term eq. (37) is obtained from eq. (39b) by taking two gauge field derivatives at $A = 0$.

A further relevant example is the sum of the classical action and the term $\Gamma_F$ in eq. (20) that generates the effective gluon mass. This combination is obtained with

$$Z_A(-D^2) = Z_A - Z_F D^2 \,, \quad f_{A,\gamma\delta\mu\nu\rho\sigma} = 0 \,. \tag{40}$$

Here, $Z_A$ is the constant background wave function renormalisation multiplying the classical action, which also entails $Z_A = Z_{g_s}^{-2}$.

The example given in eq. (40) is central for two reasons: Firstly, it demonstrates how the condensate studied in this work emerges from the general, gauge-invariant form of the effective action eq. (39a), which is defined in the next section within the background field formalism. Secondly, it establishes a link between the wave function renormalisation of the condensate and the kinetic operator of the gluon field $Z_A(\Delta_s)$. More explicitly, due to the generality of the split eq. (39b), eq. (40) entails that the wave function renormalisation of the condensate eq. (20) is simply given by the $D^2$-coefficient of the dressing function of the gluon propagator. In the limit of vanishing background, this simply corresponds to the $p^4$-term in the inverse gluon propagator.

Note that the use of different $\Delta_s$ in the split eq. (39b) leads to different forms for $f_{\mu_1 \cdots \mu_6}$, thus modifying the parametrisation of the kinetic term. Still, the different field modes carry different spin, and the use of the respective Laplacians makes the split in eq. (39b) to be the most natural. Typically, higher order terms within this split are suppressed in the effective action. For example, the second derivative of the classical Yang-Mills action with respect to the gauge field is given by $\Delta_1 = \mathcal{D}_T$, multiplied by a covariant transverse projection operator. For covariantly constant fields with $[D, F] = 0$, we get

$$\frac{\delta^2}{\delta A_\rho \delta A_\sigma} \frac{1}{2} \int_x \mathrm{tr} F_{\mu\nu}^2 = \mathcal{D}_{T,\rho\gamma} \, \Pi_{\gamma\sigma}^\perp(D) \,, \tag{41}$$

where the trace is taken in the fundamental representation. Above, we introduced the covariant transverse and longitudinal projections,

$$\Pi_{\mu\nu}^\perp(D) = \delta_{\mu\nu} - \Pi_{\mu\nu}^\parallel(D) \,, \quad \Pi_{\mu\nu}^\parallel(D) = D_\mu \frac{1}{D^2} D_\nu \,. \tag{42}$$

Equation (42) defines a decomposition in a covariantly transverse subspace with $D_\mu \Pi^\perp(D) = 0$. It is complete, $\Pi^\perp(D) + \Pi^\parallel(D) = \mathbb{1}$, and trivially orthogonal. Finally, the operators have the projection property $(\Pi^\perp(D))^2 = \Pi^\perp(D)$ and $(\Pi^\parallel(D))^2 = \Pi^\parallel(D)$.

## 3.2 Ghost and gluon two-point functions

When supplemented by a wave function renormalisation $Z_A(\mathcal{D}_T)$, eq. (41) provides a very good approximation of the full two-point function of the background gluon. This suggests the split in eq. (39b) with the spin one Laplacian $\Delta_1 = \mathcal{D}_T$ for the transverse two-point function, and with the second term being subleading,

$$\Gamma_{AA,\mu\nu}^{(2,0)}[A, 0] = Z_A(\mathcal{D}_T) \mathcal{D}_{T,\mu\sigma} \, \Pi_{\sigma\nu}^\perp(D) + F_{\gamma\delta} \, \Delta f_{A,\gamma\delta\mu\sigma}(D) \, \Pi_{\sigma\nu}^\perp(D) \,, \tag{43}$$

where $\Delta f_{A,\gamma\delta\mu\nu}$ is a combination of derivatives of $f_{A,\mu\nu\rho\sigma}$ fully contracted with powers of the field strength, see eq. (39b), and $\bar{A} = A$. The transversality of eq. (43) follows from the gauge invariance of the background field effective action, as does its covariance. In eq. (43) we have used the notation

$$\Gamma_{\bar{A}^n \phi_{i_1} \cdots \phi_{i_m}}^{(n,m)}[\bar{A}, \phi] = \frac{\Gamma[\bar{A}, \phi]}{\delta \bar{A}^n \delta \phi^m} \,, \qquad \phi = (a, c, \bar{c}) \,, \tag{44}$$

with $\phi$ denoting the ghost and gluon fluctuation field. We shall use the split eq. (39b) leading to eq. (43) and similar natural splits for the covariant versions of the momentum dependent two-point functions, thus going from the Landau gauge to the Landau-DeWitt gauge.

In particular one finds, that a similar line of arguments holds true for the kinetic operator $Z_a(p^2)p^2$ of the fluctuation field $a_\mu$,

$$\Gamma_{aa,\mu\nu}^{(0,2)}[0, 0] = Z_a(p^2) p^2 \Pi_{\mu\nu}^\perp(p) + \frac{1}{\xi} p^2 \Pi_{\mu\nu}^\parallel(p) \,, \tag{45}$$

where eq. (38) was employed, and a diagonal form in the algebra, $\mathbb{1}^{ab} = \delta^{ab}$, is implied. Background gauge invariance entails that $\Gamma^{(0,2)}[A, 0]$ is a covariant operator under the background gauge transformations eq. (35). In consequence, the transverse part of $\Gamma_{aa}^{(0,2)}[A, 0]$ can be parametrised by the generic form of a background gauge covariant function already employed in eq. (39b), i.e.,

$$\Gamma_{aa,\mu\nu}^{(0,2)}[A, 0] = Z_a(\mathcal{D}_T) \mathcal{D}_{T,\mu\sigma} \, \Pi_{\sigma\nu}^\perp(D) - \frac{1}{\xi} D^2 \, \Pi_{\mu\nu}^\parallel(D) + F_{\gamma\delta} \, \Delta f_{a,\gamma\delta\mu\sigma}(D) \, \Pi_{\sigma\nu}^\perp(D) \,. \tag{46}$$

In eq. (46) we have used the spin-1 Laplacian $\Delta_1 = \mathcal{D}_T$ defined in eq. (39c) in the wave function renormalisation $Z_a$, since the transverse fluctuating gluon is a spin-1 field. For two-flavour QCD, the validity of such covariant expansions has been confirmed explicitly for the quark-gluon vertex, whose non-classical tensor structure can be related to higher order gauge-invariant terms $\bar{q}\slashed{D}^n q$ [67].

Finally, in the case of the ghost two-point function we parametrise

$$\Gamma_{c\bar{c}}^{(0,2)}[A,0] = -D^2 Z_c(-D^2) + F_{\mu\nu}\Delta f_{c,\mu\nu}(D), \tag{47}$$

where the use of the spin zero Laplacian in eq. (47) is suggested by the ghost being a spin zero field. For $A_\mu = 0$, the ghost two point function in eq. (47) reduces to that in standard covariant gauges.

The infrared behaviour of $Z_a(p)$ in the Landau gauge is an extensively studied subject, both on the lattice as well as with functional approaches, see e.g. [26, 27, 29–31, 35]. In particular, two types of solutions have emerged:

(*i*) The *scaling* solution [71] has an infrared vanishing gluon propagator and a scaling infrared behaviour,

$$Z_{a,\mathrm{IR}} \propto (-D^2)^{-2\kappa}, \qquad Z_{c,\mathrm{IR}} \propto (-D^2)^{\kappa}, \tag{48}$$

with $\kappa \approx 0.6$. In eq. (48) we have dropped terms proportional to the field strength. Note that in this IR solution the ghost dressing function is infrared divergent. For the present computations we shall use the fRG results from [47] within a quantitatively reliable approximation, for respective DSE results see [72].

(*ii*) An entire family of *decoupling* or *massive* solutions [40], where the gluon propagator and the ghost dressing function saturate at finite non-vanishing values at the origin, in agreement with the IR behaviour found in large-volume lattice simulations. Specifically, we have

$$Z_{a,\mathrm{IR}} \propto \frac{1 + c_a D^2 \log\left(\frac{-D^2}{\Lambda_{\mathrm{QCD}}^2}\right)}{-D^2}, \qquad Z_{c,\mathrm{IR}} \propto c_c. \tag{49}$$

Note that the fluctuating propagator can be mapped to the background one by means of an exact identity, characteristic of the Batalin-Vilkoviski formalism, which involves a special two-point function, see e.g. [30, 73].

We emphasise that both types of solutions agree quantitatively for momenta $p^2 \gtrsim \Lambda_{\mathrm{QCD}}^2$, with $\Lambda_{\mathrm{QCD}}$ related to the infrared mass gap. As a result, the deviations induced to phenomenological observables by the use of either type are quantitatively minimal, see e.g. [67, 74]. In fact, in the present work we will cover all potential solutions listed above, and show that their IR differences are immaterial to the central question of the presence of dynamical condensate formation.

Both types of solutions, eq. (48) and eq. (49), are infrared irregular, and do not admit a Taylor expansion about $-D^2 = 0$. Instead, we can expand the wave function renormalisations about the infrared asymptotics. Making use of the relation between the condensate and gluon wave function renormalisation established in eq. (40), we arrive at

$$Z_{a/A}(-D^2) = Z_{a/A,\mathrm{IR}}(-D^2) + (-D^2)Z_{a/A,F} + \mathcal{O}(D^4), \tag{50}$$

for both $Z_a$ and $Z_A$ with $Z_{a/A,\mathrm{IR}}$ defined in eq. (48) and eq. (49), and $Z_{a/A,F}$ is the wave function $Z_F$ for fluctuation and background field respectively. The first term $Z_{a/A,\mathrm{IR}}$ carries the irregular

infrared asymptotic behaviour, and $Z_{a/A,F}$ is the (uniquely defined) constant prefactor of the linear term in $-D^2$. The expansion eq. (50) makes explicit that scaling and decoupling solutions only differ in the IR leading term $Z_{a/A,\text{IR}}$, while coinciding in the expansion in powers of $-D^2$. This in particular entails that the overlap between gluon propagator and the condensate eq. (20) is independent of the leading IR behaviour of the respective solution, scaling or decoupling.

We are ultimately interested in the physical mass gap $m_{\text{gap}}$ of the fluctuation field $a_\mu$ resulting from the condensate term eq. (20) in the full field $A = \bar{A} + a$. The derivation of the fluctuation field mass gap works analogously to that of eq. (32) in Section 2.3, and leads to a contribution $\Gamma_{\text{gap}}$ in the effective action with

$$\Gamma_{\text{gap}} = \frac{1}{2} m_{\text{gap}}^2 \int_x a_\mu^b a_\mu^b \,, \tag{51}$$

where the effective gluon mass of the fluctuation gluon $a_\mu$ is given by

$$m_{\text{gap}}^2 = \frac{Z_{\text{cond}}}{2} f_{\text{av}}(N_c) \langle F \rangle^2 \,, \tag{52}$$

with $Z_{\text{cond}} = Z_{a,F}$ and the averaging factor $f_{\text{av}}(N_c)$ introduced in eq. (29) and discussed there. In particular we have $Z_F = Z_{A,F} \neq Z_{\text{cond}}$. The wave function $Z_F$ is used in eq. (32) for the mass term in a gauge invariant effective action, and in the present approach this is the background field effective action. The difference between the wave functions $Z_F$ and $Z_{\text{cond}}$ is the ratio of the respective wave functions of the background and fluctuation gluons.

In eq. (40) we observed that the wave function renormalisation $Z_{\text{cond}}$ of the condensate studied here generally appears in the dressing function of the respective gluon propagator, cf. eq. (50). This connection will be utilised in Section 5 to determine $Z_{\text{cond}}$ from the input gluon propagators [47] employed in the computation of the background field effective potential $\mathcal{W}_{\text{eff}}(F^a)$. Supplemented with the non-trivial effective potential minimum $\langle F \rangle$, this procedure eventually lead to our heuristic estimate of the gluon mass gap in Landau gauge Yang-Mills theory.

### 3.3 Large $N_c$-scaling and self-consistency

The effective gluon masses $m_A^2$ in eq. (32) and $m_{\text{gap}}^2$ in eq. (52) show an explicit $1/N_c$-scaling, while no $N_c$-scaling is present in the large $N_c$ limit, if the theory is formulated in the 't Hooft coupling

$$\lambda = N_c g_s^2 \,. \tag{53}$$

This property serves as a self-consistency check of our computation and specifically our group average used to derive eq. (32), eq. (52) and entailed in $f_{\text{av}}(N_c)$

An illustrative and relevant example are the functional relations of the two-point function $\Gamma_{aa}^{(0,2)}(p)$. Cast in a relation for the wave function $Z_a(p)$, they read

$$Z(p^2) = Z_{\text{in}} + g_s^2 N_c \, \text{Diags}_1 + \mathcal{O}(N_c^0) \,, \tag{54}$$

where the right hand side stands for the typical loop diagrams of e.g. (integrated) fRG flows or Dyson-Schwinger equations. Here, $Z_{\text{in}}$ stands for the input dressing, either the one at the initial UV cutoff scale (fRG) or the classical dressing (DSE). In most cases the $\mathcal{O}(N_c^0)$ term is dropped, for an exception as well as a respective discussion see [75]. The term $\text{Diags}_1$ stands for the loop integral that depends on the wave functions of all the fields and the full vertex dressings. Importantly, the functional relations for all other vertex dressings and wave functions have the

same form as eq. (54). Accordingly, if dropping the subleading term of the order $O(N_c^0)$, all functional relations only depend on the 't Hooft coupling eq. (56), and so do all correlation functions. Respective lattice studies also reveal that the large $N_c$-limit is achieved already for $N_c \gtrsim 3$ for most correlation functions, for a review see [76].

In summary we deduce, that in the large $N_c$-limit the only $N_c$-dependence of the effective gluon masses $m_A^2$ in eq. (32) and $m_{\text{gap}}^2$ in eq. (52) is implicit in the dependence on the 't Hooft coupling eq. (56). This concludes our brief discussion of the $N_c$-scaling of correlation functions.

The relations for the effective gluon mass, eq. (32), eq. (52), show an even more direct scaling consistency: $Z_{\text{F}}$ is an expansion term in the two-point function of the fluctuating gluon. Moreover, in the presence of the condensate this two-point function approaches the effective gluon for vanishing momentum,

$$\lim_{p \to 0} \Pi_{\mu\nu}^{\perp}(p) \Gamma_{aa,\mu\nu}^{(0,2)}(p) = 3 m_{\text{gap}}^2 . \tag{55}$$

Accordingly, both $Z_{\text{cond}}$ and $m_{\text{gap}}$ have the same $N_c$-scaling (only dependent on the 't Hooft coupling in the large $N_c$-limit) as well as the same RG-scaling. In conclusion, the ratio $Z_{\text{cond}}/m_{\text{gap}}^2$ is manifestly RG-invariant as well as $N_c$-independent in the large $N_c$-limit. This implies already, that the RG-invariant information in the effective gluon mass is given by $f_{\text{av}}(N_c) \langle F \rangle^2$. The value of the mass itself depends on the RG-condition and should not be confused with the gluon mass gap. The latter can be defined as the inverse screening length of the gluon propagator which is indeed RG-invariant.

In summary, $f_{\text{av}}(N_c) \langle F \rangle^2$ should be $N_c$-independent in the large $N_c$-limit. This fixes the $N_c$-scaling of $f_{\text{av}}(N_c)$, given that of $\langle F \rangle^2$. The $N_c$-scaling of the latter is obtained by an $N_c$-analysis of the effective potential, whose explicit computation is detailed in Section 4 and Appendix D. Here we only need that it consists out of an ultraviolet classical piece of the form eq. (12) and a term that depends on $N_c F^2$,

$$\mathcal{W}_{\text{eff}}(F^a) = \frac{1}{4g_s^2} F^2 + \Delta \mathcal{W}_{\text{eff}}(N_c F^2) , \tag{56}$$

see Section 4.2. In eq. (56), $g_s^2$ is the strong coupling at a large momentum scale $k_{\text{UV}}$, and we will use $k_{\text{UV}} = 20$ GeV for this scale later on. We now absorb $N_c$ into the field strength squared amplitude $F^2$, i.e. $\bar{F}^2 = N_c F^2$. With eq. (53) this leads us to

$$\mathcal{W}_{\text{eff}}(F^a) = \frac{1}{4\lambda} \bar{F}^2 + \Delta \mathcal{W}_{\text{eff}}(\bar{F}^2) , \tag{57}$$

and consequently

$$\langle \bar{F} \rangle = \bar{F}_{\text{min}}(\lambda) \qquad \longrightarrow \qquad \langle F \rangle = \frac{1}{\sqrt{N_c}} \bar{F}_{\text{min}}(\lambda) . \tag{58}$$

The $1/N_c$-scaling for $\langle F \rangle^2$ derived in eq. (58), is confirmed numerically in Appendix D. There, the effective potential and its minimum is computed in a leading order $N_c$ approximation and hence shows the asymptotic $1/N_c$ scaling even for $N_c = 2$. This $N_c$-scaling is rooted in the adjoint representation trace of $n^a t^a$ appearing the definition of the covariantly constant field strength in eq. (10), cf. eq. (88). We have confirmed its numerical presence in a comparison of $N_c = 2, 3$.

## 4 Background field effective potential

Now we compute the value of the field strength condensate $\langle F_{\mu\nu} \rangle$ discussed in Section 2.2. For this purpose, we update the fRG computation done in [46] to a self-consistent one with

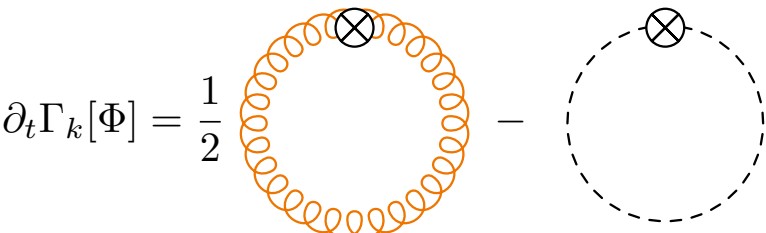

Figure 2: Depiction of the flow equation for the effective action, eq. (59). Spiralling orange lines depict the full field-dependent gluon propagator $\langle AA \rangle_c = G_{aa}[\bar{A}, \phi]$, dashed back lines depicted the full field-dependent ghost propagator $\langle c\bar{c} \rangle_c = G_{c\bar{c}}[\bar{A}, \phi]$, where the subscript stands for connected part. The circled cross stands for the regulator insertions $\partial_t R_a$ (gluon loop) and $\partial_t R_c$ (ghost loop).

fRG precision gluon and ghost propagators from [47]. In Section 4.1 we briefly review the approach, and in Section 4.3 we report on the results for the condensate.

## 4.1 Flow of the background field effective potential

For the full computation we resort to the functional renormalisation group approach, for QCD-related reviews see [33–35, 55, 77, 78]. In this approach, an infrared regulator $R_k(p)$ is added to the classical dispersion. In the infrared, that is $p/k \to 0$, the regulator endows all fields with a mass, typically proportional to the cutoff scale $k$. In addition, the regulator $R_k(p)$ vanishes rapidly as $p/k \to \infty$, and the ultraviolet physics is not modified. The change of the scale dependent effective action, $\Gamma_k$, under a variation of the cutoff scale $k$ is described by the flow equation. In the background field approach it reads

$$\partial_t \Gamma_k[\bar{A}, \phi] = \frac{1}{2} \operatorname{Tr} R_a[\bar{A}] G_{aa}[\bar{A}, \phi] - \operatorname{Tr} R_c[\bar{A}] G_{c\bar{c}}[\bar{A}, \phi], \tag{59}$$

where $t = \log k/\Lambda$ is the (negative) RG time, and $G_A, G_c$ are the fluctuation propagators of gluon and ghost respectively,

$$G_{\phi_1 \phi_2}[\bar{A}, \phi] = \left[ \frac{1}{\Gamma_k^{(0,2)}[\bar{A}, \phi] + R_k[\bar{A}]} \right]_{\phi_1 \phi_2}. \tag{60}$$

The traces in eq. (59) sum over momenta, Lorentz and gauge group indices, details can be found in Appendix C. The regulator function $R_k = (R_a, R_c)$ transforms covariantly under background gauge transformations, which preserve the background gauge invariance of the effective action. The current work utilises the propagator data from [47], which requires the use of the same regulators for our computation of the background field effective potential. For details on the regulators see Appendix B.

For the derivation of the (background) field strength condensate we solve the equation of motion stemming from the effective potential $\mathcal{W}_{\text{eff}}(F^a)$ of covariantly constant field strength defined in eq. (9). In the fRG approach it is obtained from its scale-dependent analogue,

$$\mathcal{W}_k(F^a) = \frac{1}{\mathcal{V}} \Gamma_k[A(F^a), 0], \tag{61a}$$

with the full effective potential being defined at vanishing cutoff scale $k = 0$,

$$\mathcal{W}_{\text{eff}}(F^a) = \mathcal{W}_{k=0}(F^a). \tag{61b}$$

The effective potential $\mathcal{W}_k$ is obtained by integrating the flow equation of the background field effective action $\partial_t \Gamma_k[A(F), 0]$, derived from eq. (59) from the initial ultraviolet scale $k_{\text{UV}}$ to the running cutoff scale $k$. The only input in this flow are the two-point functions $\Gamma_{aa}^{(0,2)}[A(F), 0]$ and $\Gamma_{c\bar{c}}^{(0,2)}[A(F), 0]$, which we can infer from Landau gauge results. This is the background Landau-deWitt gauge with $\bar{A} = 0$. For vanishing background the two-point functions only depend on momenta, $\Gamma_k^{(0,2)}(p)$. We use the results from [47], with

$$\Gamma_{aa,k}^{(0,2)}(p) = p^2 Z_{a,k}(p^2) \Pi^\perp(p) + p^2 \left[ \frac{1}{\xi} + Z_{a,k}^\parallel(p^2) \right] \Pi^\parallel(p),$$

$$\Gamma_{c\bar{c},k}^{(0,2)}(p) = p^2 Z_{c,k}(p^2), \tag{62}$$

with the transverse and longitudinal projection operators introduced in eq. (38). In eq. (62), $\mathbb{1}^{ab} = \delta^{ab}$ is implied in both two-point functions. The longitudinal dressing $Z_{a,k}^\parallel$ signals the breaking of BRST invariance due to the presence of the regulators, and vanishes in the limit $k \to 0$. There, the gluon two-point function in eq. (62) reduces to that of eq. (45). Moreover, $Z_{a,k}^\parallel$ is absent in the gluon propagator for the Landau gauge, $\xi \to 0$,

Now we switch on the background field and use the decomposition eq. (46) for the transverse gluon two-point function. In addition, we drop the second line proportional to $\Delta f_a$ comprising higher order terms. They are associated with non-classical tensor structures and can be shown to be small in the perturbative and semi-perturbative regimes. In the Landau-DeWitt gauge, only the gauge-fixing survives in the longitudinal propagator and we can drop the cutoff contribution $Z_{a,k}^\parallel$. For the ghost we use eq. (47), where we drop the second term proportional to $\Delta f_c$. This leads us to

$$\Gamma_{aa,k}^{(0,2)}(p) \simeq \mathcal{D}_T Z_{a,k}(\mathcal{D}_T) \Pi^\perp(-D) - \frac{1}{\xi} D_\mu D_\nu,$$

$$\Gamma_{c\bar{c},k}^{(0,2)}(p) \simeq -D^2 Z_{c,k}(-D^2), \tag{63}$$

valid for covariantly constant field strength with $[D, F] = 0$. For these configurations, the transverse projection operator commutes with functions of the Laplacians $\Delta_0$ and $\Delta_1$.

## 4.2 RG-consistent initial condition

The flow equation eq. (61a) of the effective potential $\mathcal{W}_k(F^a)$ is readily obtained by inserting the approximations of eq. (63) into the flow eq. (59). The flow is evaluated for the generic condensate background eq. (10). The details can be found in Appendix B. Finally, the effective potential $\mathcal{W}_{\text{eff}}(F^a)$ of Yang-Mills theory is obtained from the integrated flow. We arrive at

$$\mathcal{W}_k(F^a) = \mathcal{W}_{k_{\text{UV}}}(F^a) + \int_{k_{\text{UV}}}^k \frac{\text{d}k'}{k'} \partial_{t'} \mathcal{W}_{k'}(F^a), \tag{64}$$

where $\mathcal{W}_{k_{\text{UV}}}$ is well approximated by the classical potential eq. (12) for a large initial cutoff scale $k_{\text{UV}}$. Perturbation theory is valid for these scales, and the background field effective action $\Gamma_{k_{\text{UV}}}[A]$ reduces to the classical Yang-Mills action of eq. (1), augmented with a wave function renormalisation $Z_{A,k_{\text{UV}}}$. All other terms are suppressed by inverse powers of $k_{\text{UV}}$. This amounts to

$$\mathcal{W}_{k_{\text{UV}}}(F^a) = \frac{Z_{A,k_{\text{UV}}}}{4 g_s^2} F^2 = \frac{F^2}{16 \pi \alpha_s(k_{\text{UV}})}, \tag{65}$$

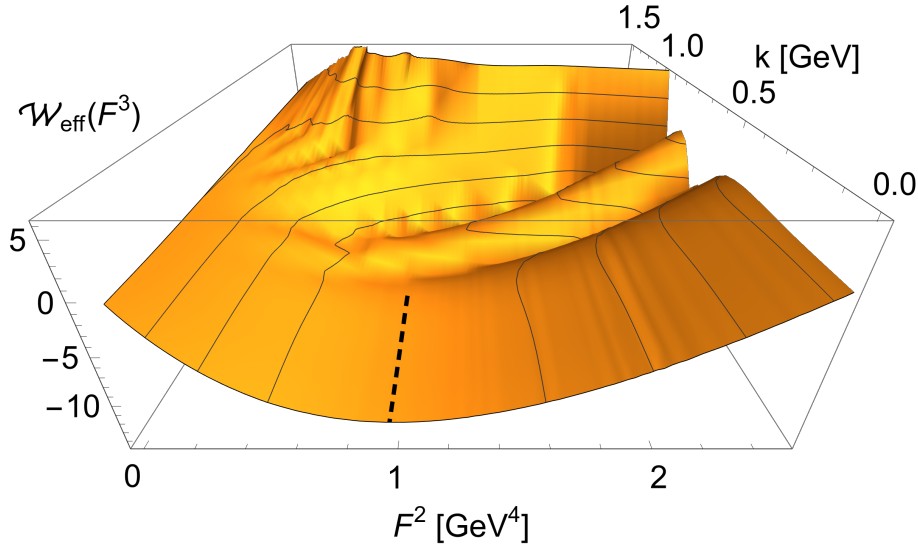

Figure 3: Effective Potential as a function of $F^2$ with a field strength pointing in the $t^3$-direction, $(n^3, n^8) = (1, 0)$, and the cutoff scale $k$. The dashed line singles out the absolute minimum of $W(F)$, see eq. (68). The substructure of the potential at cutoff scale $k \gtrsim 0.5$ GeV is related to the regulator used, see Appendix D. It leaves no trace in the potential for $k \to 0$.

where

$$\alpha_s(k) = \frac{1}{4\pi} \frac{g_s^2}{Z_{A,k}}, \quad \text{with} \quad Z_{A,k_{\mathrm{UV}}} = 1. \tag{66}$$

Here, $Z_{A,k}$ is the background wave function $Z_{A,k}(p = 0)$, and $g_s^2$ is the running coupling at the initial scale $k_{\mathrm{UV}}$.

The onset of this asymptotic UV regime for cutoff scales $k \gtrsim k_{\mathrm{on}}$ depends on the chosen regulator or rather its shape. Roughly speaking, the sharper the regulator drops of in momenta at about the cutoff scale, the larger is the onset scale $k_{\mathrm{on}}$. For the ghost and gluon regulators underlying the computation of the propagators in [47], eq. (83), we choose an initial scale $k_{\mathrm{UV}} = 20$ GeV. This is safely in the asymptotic UV regime of the regulators eq. (83), as is also explicitly discussed in Appendix D. In summary, the computation is initialised at

$$\alpha_s(k_{\mathrm{UV}}) = 0.184, \quad \text{with} \quad k_{\mathrm{UV}} = 20 \text{ GeV}, \tag{67}$$

and the running coupling data are also taken from [47], which ensures the self-consistency of the computation.

In eq. (66) we have used that the background wave function renormalisation $Z_A$ satisfies $Z_A^{-1} = Z_{g_s}^2$, a consequence of background gauge invariance. Moreover, RG-consistency, see e.g. [55, 79], enforces eq. (66): the flow of the initial effective action with an infinitesimal change of the initial cutoff scale is given by the flow equation. Phrased in terms of the effective potential in eq. (64), this is the simple requirement that $\mathcal{W}_k$ and in particular $\mathcal{W}_{\mathrm{eff}} = \mathcal{W}_k$ is independent of $k_{\mathrm{UV}}$. Then, differentiation of eq. (64) with respect to $k_{\mathrm{UV}}$ readily leads to eq. (65). More details are deferred to Appendix D.

## 4.3 Results

The above derivation allows the numerical computation of the scale dependent effective potential $\mathcal{W}_k(F^a)$ by performing the integration in eq. (64) up to the respective RG-scale $k$. The

result is shown in Figure 3, which shows the $k$-dependent effective potential as a function of $F^2$, with a field strength pointing in the $t^3$-direction: $(n^3, n^8) = (1, 0)$. The condensate $\langle F \rangle$ is given by the solution of the equation of motion (EoM) for the effective potential $\mathcal{W}_{\text{eff}}(F^a)$, given by

$$\frac{\partial \mathcal{W}_{\text{eff}}(F^a)}{\partial F}\bigg|_{F=\langle F \rangle} = 0, \tag{68}$$

for the generic field strengths of eq. (15). The emergence of a non-trivial minimum is clearly visible in the non-perturbative regime $\lesssim 1$ GeV, and its position indicated with the black dashed line in Figure 3.

The gauge invariant information of the field strength $F_{\mu\nu}$ is stored in its eigenvalues, which do not change under (unitary) gauge transformations. In the present case, only the $F_{01} = F_{23}$ components and their anti-symmetric counterparts are non-vanishing, and they are proportional to a combination of the Cartan generators, see eq. (15). The traces in the flow equation are in the adjoint representation, and the six non-vanishing eigenvalues of $n^3 t^3 + n^8 t^8$ are given by

$$\tau_{\pm}^{(1)} = \pm\, n^3,$$

$$\tau_{\pm}^{(2)} = \pm\left(\frac{1}{2}n^3 + \frac{\sqrt{3}}{2}n^8\right),$$

$$\tau_{\pm}^{(3)} = \pm\left(\frac{1}{2}n^3 - \frac{\sqrt{3}}{2}n^8\right), \tag{69}$$

for more details see e.g. [80, 81]. The global, degenerate minima in Figure 1 are located in the direction of the eigenvectors. The underlying Weyl symmetry maps the different minima into each other, and is seen in Figure 1.

From eq. (68) we determine the expectation values or rather saddle point position of the condensate in both directions. We find that the expectation value in $n^3$-direction is a global minimum, while in the $n^8$-direction the EoM singles out a saddle point. Both points are indicated by the red and blue dots respectively in Figure 1. We determine the value of the minimum by interpolation,

$$\langle F \rangle_{\lambda_3}^2 = 0.98(11)\,\text{GeV}^4, \tag{70}$$

where the error is obtained by a variation of 2% in the initial coupling $\alpha_s$. More details on the RG-consistency of this procedure are provided in Appendix D. Equation (70) is the result of an $SU(3)$ computation without the $N_c$ rescaling.

As discussed below eq. (15), the minimum in eq. (70) is composed by the condensates of both $F^2$ and $F\tilde{F}$. Due to the CP-violating nature of an $F\tilde{F}$ condensate, its contribution to our condensate value is tightly constrained by experimental data. Nonetheless, the value quoted in eq. (70) should be interpreted as an estimate colorless condensate $\langle F^2 \rangle$.

The present first-principle Yang-Mills result eq. (70) corroborates the phenomenological estimates, i.e. $\langle F^2 \rangle = 0.854(16)\,\text{GeV}^4$ [82], as already remarked in [46]. Indeed, the normalisation procedure used here is similar to that in the phenomenological computation. In contrast, both eq. (70) and the phenomenological estimates disagree with the lattice estimate $\langle F^2 \rangle = 3.0(3)\,\text{GeV}^4$ [83]. The latter value is extracted from $\langle G^2 \rangle = 0.077(7)$ in [83], and applying $\langle F^2 \rangle = 4\pi^2 \langle G^2 \rangle$. In this context we remark that the total normalisation may differ, even though all procedures provide RG-invariant results: for example, one may multiply the respective result by the RG-invariant ratio of couplings at different momenta, $\alpha_s(p_1^2)/\alpha_s(p_2^2)$, resulting in a global factor. This amounts to mapping the factor $\alpha_s$ from one momentum scale

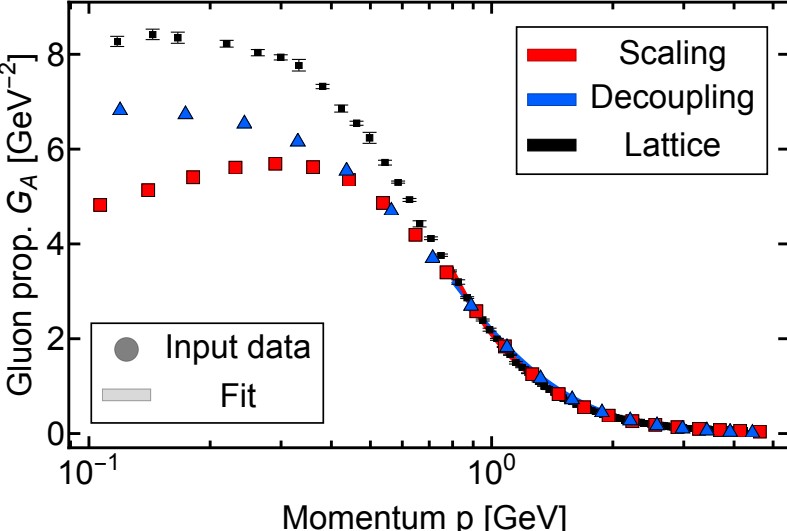

Figure 4: Gluon propagators from the fRG [47] in the scaling (red) and decoupling (blue) scenario as well as lattice data from [22] with a continuum and infinite volume extrapolation, see [84,85]. Coloured/black markers show the data. Solid lines show the respective fits from which the wave function renormalisation $Z_{\text{cond}}$ (cf. eq. (72)) is computed. The fit Ansätze are given in eq. (95). Here, we plot fits to the propagator data over the maximal fit interval, see also Appendix E for details.

to another. While we lack a comprehensive interpretation, we simply point out that the lattice definition involves $\alpha_s$ at a low momentum scale, conversely to the present procedure, and that used in phenomenological applications.

For comparison we also provide the saddle point value,

$$\langle F \rangle^2_{\lambda_8} = 0.85(11)\,\text{GeV}^4\,, \tag{71}$$

which may be used for a further error estimate of the relation between octet and colorless condensates, as the octet condensate should be averaged over all color directions.

## 5 Gluon mass gap

The aim of this section is to use eq. (52) and eq. (70) for an estimate of the mass gap. Evidently, to accomplish this, the determination of the wave function renormalisation $Z_{\text{cond}}$ is required.

Inspecting the condensate generating kinetic term, see eq. (20), one finds that its analogue for the fluctuating gluon also contains contributions of the type

$$\frac{Z_{\text{cond}}}{2} \int_x a^a_\mu (\partial^2)^2 \Pi^\perp_{\mu\nu}(\partial) a^a_\nu + \dots . \tag{72}$$

Hence, the kinetic term for the field strength not only gives rise to the condensate, but also overlaps with the gluon propagator. More specifically, as can be read off eq. (72), the $p^4$-term of the fluctuation gluon two-point function carries the wave function renormalisation $Z_{\text{cond}}$ as a prefactor, as made explicit in eq. (50).

Note that by means of eq. (39a) and eq. (40), the $p^4$-term must be solely given by eq. (72), as $Z_A$ implicitly defined in eq. (39b) encodes the full gluon propagator dressing function, see eq. (37). In terms of an operator product expansion, $Z_{\text{cond}}$ can be extracted by determining the $p^4$-coefficient in the origin of the inverse input gluon propagator data from [47],

used in the calculation of the condensate effective potential in Section 4. This is done via a fit, given by

$$Z_{\text{fit}}(p^2) = Z_{\text{as}}(p^2) + Z_{p^2} + Z_{\text{cond}}\, p^2\,, \tag{73}$$

where only the infrared asymptotes $Z_{\text{as}}(p^2)$ distinguish between scaling and decoupling solutions (cf. eq. (48) and eq. (49). A detailed discussion of the fitting procedure is provided in Appendix E, and the respective fits in comparison to the propagator data from [47] and the lattice data of [22] are depicted in Figure 4.

Equation (73) makes it apparent that scaling and decoupling solutions differ only in the infrared, where the $p^4$-term is subleading. We determine $Z_{\text{cond}}$ from the fRG scaling solution of [47] as well as the lattice decoupling solution of [22]. Combining both estimates, we arrive at the value for the wave function renormalisation

$$Z_{\text{cond}} = 0.149(19)\,\text{GeV}^{-2}\,. \tag{74}$$

Now we use the wave function renormalisation from eq. (74), the condensate value $\langle F^2 \rangle$ eq. (33) as well as the saturation bound eq. (33) for the averaging factor $f_{\text{av}}$ in the relation for the effective gluon mass eq. (52). This leads us to

$$m_{\text{gap}} = 0.312(27)\,\text{GeV}\,. \tag{75}$$

Equation (75) is the main result of the present work and provides an estimate for the effective gluon mass in the Landau gauge. The relatively large uncertainty in eq. (75) originates predominantly from the error for $Z_{\text{cond}}$ in eq. (74). In particular, it does not include a systematic error estimate, and is solely rooted in the small amount of data points for the gluon propagator of [47] in the deep IR.

A large source for the systematic error is the current lack of a quantitative color average as discussed in detail in Appendix A. Moreover, the field strength condensate eq. (70) also receives contributions from the topological condensate $\langle F\tilde{F} \rangle$, see the discussion there and below eq. (15). Accordingly, we simply note that inserting the literature value from phenomenological $\langle F^2 \rangle$ estimates [82] reduces the value in eq. (75) to $m_{\text{gap}} = 0.291(19)\,\text{GeV}$. The same value is obtained by the use of the saddle point value eq. (71), which we use as an error estimate.

We can compare our result for the effective gluon mass eq. (75) with that deduced from the lattice data [22] with a continuum and infinite volume extrapolation, see [84, 85]. These data are shown in Figure 4, and the mass gap is given by the value of the inverse lattice propagator in the origin. We find

$$m_{\text{gap}}^{(\text{lattice})} = 0.3536(11)\,\text{GeV}\,, \tag{76}$$

which agrees within two standard deviations with our estimate eq. (75).

A further direct test of the present results is provided by the comparison with the effective gluon mass in eq. (104) obtained via the Schwinger mechanism with $m_{\text{gap}} = 0.320(35)\,\text{GeV}$ after scale matching. This is an alternative approach for the dynamical emergence of a gluon mass gap in the Landau gauge, for details see Appendix F. The results compare very well, which is to be expected as our propagator with the gluon mass gap agrees well with the lattice results, as does the propagator obtained with the Schwinger mechanism.

We emphasise that the estimate for the gluon mass gap depends on our choice for the color averaging factor $f_{\text{av}}$ in eq. (29): with eq. (33) we have saturated the 'natural' bound $c_{\text{av}} = 1$ in eq. (31), leading to eq. (34). In fact, the non-trivial compatibility of the present results with that obtained from lattice propagators and via the Schwinger mechanism corroborates the aforementioned choice.

We close this section with the remark that, while the effective gluon mass or rather the gluon mass gap in the Landau or Landau-DeWitt gauge is a gauge variant quantity, its size is directly related to physical scales such as the string tension and the confinement-deconfinement temperature, see [36, 37]. Still, its value varies with the gauge as does its precise relation to the physical scales and mechanisms. Consequently, the numerical estimates of its value are rather disparate, ranging from a few hundred MeV up to 1 GeV, depending on the details of the approach and the definition employed, see e.g. [8, 25, 40, 86–98]. Nonetheless, all these determinations convey information about the same gauge-invariant physical information, namely the Yang-Mills mass gap.

## 6 Summary and outlook

In the present work we have explored the dynamical emergence of a mass gap in the Yang-Mills correlation functions via the formation of color condensates, in the physical case with the $SU(3)$ gauge group one of these condensates is the octet condensate, see eq. (4). Such a condensate may be triggered by a Higgs-type mechanism in low energy QCD, similar and potentially related to dynamical chiral symmetry breaking in QCD with the pion as pseudo-Goldstone bosons.

In the current work we have carried out a qualitative analysis within the fRG approach to QCD by computing the minimum $\langle F \rangle$ of the effective potential $W(F^a)$ in the three direction of the Cartan subgroup. This non-vanishing field strength is related to non-vanishing color condensates as discussed in Section 2.2. We have computed the effective potential $\mathcal{W}(F^a)$ for covariantly constant field strength which develops a non-trivial minimum if quantum fluctuations are successively taken into account with the fRG flow, see Figure 1. The condensate value eq. (70) is in good agreement with phenomenological estimates, but both disagree with lattice results. As discussed in section 4.3, this latter discrepancy may be due to a difference in the normalisations employed.

The relation between the gluon condensate and the mass gap is given by eq. (52). We emphasise that the mass gap eq. (52) triggered by the condensate depends on the RG-condition and naturally has the RG-properties of a mass function: while the condensate itself is independent of the RG-condition, the condensate wave function is not and carries the RG-properties of the inverse gluon propagator. Consequently, the mass gap derived from eq. (52) has the RG scaling of the inverse gluon propagator, as it should. Accordingly, for a comparison of the results for the mass gap obtained here with that in the literature the potentially different RG-schemes and conditions have to be taken into account. Most fRG-computations including the present one are done in MOM$^2$, for a detailed discussion see [74].

These considerations result in our estimate of the gluon mass gap, $m_{\text{gap}} = 0.312(27)\,\text{GeV}$, where our choice eq. (29) for the color averaging factor $f_{\text{av}}$ saturates the 'natural' bound, see also the discussion below eq. (32). This estimate compares well to the lattice estimate $m_{\text{gap}}^{(\text{lattice})} = 0.3536(11)\,\text{GeV}$. The latter values is obtained from the continuum and infinite volume extrapolation [84] of the lattice data in [22], after matching the momentum scales and the renormalisation point.

We have also compared our result for the mass gap with that obtained with the longitudinal Schwinger mechanism within the framework of the pinch technique [30], see Appendix F and the very recent analysis see [99]. This analysis leads to $m_{\text{gap}}^{(\text{Schwinger})} = 0.320(35)\,\text{GeV}$, which is in excellent agreement with our estimate.

In summary, the findings of the present work suggest that the gluon condensation as a mechanism for mass generation works well. Beyond improving the systematic error of the numerical estimate, on theoretical grounds it would be desirable to establish a deeper connection

between the Schwinger mechanism and the condensate formation.

Currently, we are upgrading the present computation with the dynamical inclusion of the composite octet condensate operator, discussed in Section 2.1. Then, the octet condensate is taken into account as an effective low energy degree of freedom, allowing us to study the relevance of a potentially non-trivial condensate dynamics. We hope to report on respective results in the near future.

We thank A. C. Aguilar, M. N. Ferreira, C. S. Schneider and N. Wink for discussions. A. W. thanks the ITP Heidelberg for its hospitality, and EMMI, Conacyt and CIC-UMSNH for support. This work is done within the fQCD collaboration [100], and is supported by EMMI and the Studienstiftung des deutschen Volkes, It is part of and supported by the DFG Collaborative Research Centre SFB 1225 (ISOQUANT) as well as by the DFG under Germany's Excellence Strategy EXC - 2181/1 - 390900948 (the Heidelberg Excellence Cluster STRUCTURES). This work is also supported by the Spanish AEI-MICINN grant PID2020-113334GB, and the grant Prometeo/2019/087 of the Generalitat Valenciana.

# A  Expansions around condensates and color averages

In this Appendix we discuss the implementation of expansions around non-trivial condensates, and comment on the subtleties of the color-averaging procedure associated with the central mass formula in eq. (34). In order to illustrates the properties and subtleties, we employ two simple examples: spontaneous symmetry breaking in a scalar $O(N)$ theory, and (color) center symmetry breaking in finite temperature Yang-Mills theory.

Let us first consider a scalar field theory with an $O(N)$ field $\phi$ (including the discrete $Z_2$ symmetry when $N = 1$ ) in the symmetric phase. In the symmetric phase, both the effective action, $\Gamma[\phi]$, as well as expectation values of observables, are typically expanded around $\phi = \phi_0$, where

$$\phi_0^2 = \lim_{\mathcal{V} \to \infty} \frac{1}{\mathcal{V}} \int_{\mathcal{V}} \langle \phi(x)\phi(0) \rangle, \tag{77}$$

is defined by the order parameter of the theory. The order parameter eq. (77) can also be obtained from

$$\phi_0 = \lim_{J \to 0} \langle \phi \rangle, \tag{78}$$

where $J$ indicates an external current (or magnetisation) coupled to the field, $\lim_{J \to 0} \int_x J\phi$, which is finally removed. Alternatively, within a finite volume one may use boundary conditions that break the symmetry, and then take the infinite volume limit.

Either way, the effective action $\Gamma$ is invariant under the full symmetry group of the underlying theory by definition, whereas the vacuum state (the solution of the equations of motion) breaks the symmetry.

Thus, quite importantly, the apparent symmetry breaking in $\Gamma$, seemingly induced by the expansion point, is absent for the full effective action. In turn, a given approximation scheme may break this symmetry (for example a finite order of a Taylor expansion about $\phi = \phi_0$). This symmetry can be restored subsequently by averaging the approximated effective action $\Gamma_{\text{app}}[\phi]$ over the symmetry group, $\Gamma[\phi] = \langle \Gamma_{\text{app}}[\phi] \rangle_{\text{av}}$. Note in this context, that in our example case of an $O(N)$ theory the averaged expectation value of the field vanishes, $\langle \phi \rangle_{\text{av}} = 0$, as it must. Moreover, the operator in eq. (77) has the full symmetry and hence does not change

under the averaging procedure, while $\langle\phi\rangle$ does.

In the case of the effective gluon mass, the underlying symmetry is a gauge-symmetry. For this reason we also consider a second, closer, example, the expectation value of the Polyakov loop $\langle L\rangle$ in finite temperature Yang-Mills theory,

$$L = \frac{1}{N_c}\operatorname{tr}\mathcal{P}\exp\{ig_s\oint A_0(x)\}, \tag{79}$$

where the integral $\oint$ in eq. (79) is over $x_0 \in [0, 1/T]$, and the trace is taken in the fundamental representation. Here, $T$ denotes the temperature and $\mathcal{P}$ is the path ordering operator. The underlying symmetry is the center symmetry $Z_{N_c}$ of the gauge group with $L \to z L$ and $z \in Z_{N_c}$. We have the order parameter

$$L_0^2 = \lim_{\mathcal{V}\to\infty}\frac{1}{\mathcal{V}}\int_{\mathcal{V}}\langle L(0)L^{\dagger}(x)\rangle, \tag{80}$$

which is non-vanishing in the confining disordered low temperature phase. Typically, both in functional approaches as well as on the lattice, eq. (80) is obtained by an infinitesimal explicit center symmetry breaking in the Cartan direction $t^3$, similar to introducing an infinitesimal explicit breaking of $O(N)$ symmetry described above. In the $t^3$ direction the Polyakov loop takes real values and we get

$$L_0 = \langle L(x)\rangle, \tag{81}$$

with a real positive $L_0$, which is a non-trivial solution of the equation of motion (of $A_0$) at finite temperature. The expectation value of the order parameter serves as a physical expansion point for observables as well as the effective action in functional approaches, both in first principle QCD computation and low energy effective theories of QCD. In quantitative approximations the results for observables agree very well with lattice simulations, for the Polyakov loop itself see [81]: The observables are either color blind in the first place and hence do not require a color average and are insensitive to it, or, as in the case of the Polyakov loop, a color direction was singled out for the computation in the first place.

However, the comparison of gauge fixed correlation functions or parts of it is more intricate, as then the averaging is required and may also affect the gauge fixing, for more details and further literature see in particular [67, 101] and the recent review [35]. This intricacy also applies in the present situation and makes a direct comparison of the effective gluon mass difficult.

The lack of a quantitative averaging procedure has forced us to introduce the averaging factor $f_{\mathrm{av}}(N_c)$ in our results, see eq. (29) and the definition of the effective gluon mass, eq. (32) and eq. (52). In the present work we have only determined its $N_c$-dependence with the consistency of the large $N_c$ scaling. As mentioned in the main text, the value of $f_{\mathrm{av}}(N_c)$ is the largest source of systematic error for the effective gluon mass.

## B  Flow of the effective potential

Here we provide some details of the computation of the integrated flow eq. (64) of the effective potential, eq. (61a) from the flow equation eq. (59) and the propagators eq. (62). Inserting

the latter into eq. (59) yields,

$$
\begin{aligned}
\partial_t \mathcal{W}_k(F^a) =& \frac{3}{2} \operatorname{Tr} \frac{\partial_t R_a^\perp(\mathcal{D}_T)}{\mathcal{D}_T Z_{a,k}(\mathcal{D}_T) + R_a^\perp(\mathcal{D}_T)} + \frac{1}{2} \operatorname{Tr} \frac{\partial_t R_a^\|(-D^2)}{-D^2 + R_a^\|(-D^2)} \\
&+ \frac{1}{2} \operatorname{Tr} P_0 \frac{\partial_t R_a^\perp(-D^2)}{-D^2 Z_{a,k}(-D^2) + R_a^\perp(-D^2)} - \operatorname{Tr} \frac{\partial_t R_c(-D^2)}{-D^2 Z_{c,k}(-D^2) + R_{k,c}(-D^2)} \\
&- \frac{3}{2} \operatorname{Tr} \frac{\partial_t R_a^\perp(p^2)}{p^2 Z_a(p^2) + R_a^\perp(p^2)} - \frac{1}{2} \operatorname{Tr} \frac{\partial_t R_a^\|(p^2)}{p^2 Z_a(p^2) + R_a^\|(p^2)} - \operatorname{Tr} \frac{\partial_t R_c(p^2)}{p^2 Z_{c,k}(p^2) + R_{k,c}(p^2)},
\end{aligned}
\tag{82}
$$

where the contributions in the first line are the glue contributions, and $P_0$ denotes the projection on the zero-mode. The traces in eq. (82) sum over momenta or space-time, as well as internal and Lorentz indices of the respective field modes. We have three covariant transverse modes and one covariant longitudinal mode, the trivial gauge mode. The term in the second line is the ghost contribution, and the field-independent subtraction in the third line normalises the potential to $\mathcal{W}_k(F^a = 0) = 0$. We choose the regulator in consistency with the input data. The regulators in [47] are defined as,

$$
\begin{aligned}
R_{a,k}(p) &= p^2 r(x) \left( \tilde{Z}_{a,k} \Pi^\perp(p) + \Pi^\|(p) \right), \\
R_{a,k}(p) &= p^2 r(x) \tilde{Z}_{c,a},
\end{aligned}
\tag{83}
$$

with the projection operators $\Pi^{\perp,\|}$ defined in eq. (38). In eq. (83), $x$ is the dimensionless momentum variable, $x = p^2/k^2$, and the shape function $r(x)$ used in [47] is given by,

$$
r(x) = \left( \frac{1}{x} - 1 \right) \frac{1}{1 + e^{\frac{x-1}{a}}}, \qquad a = 2 \times 10^{-2}.
\tag{84}
$$

The shape function eq. (84) is a smoothened version of the Litim shape function, [102]. The cutoff dependent prefactors $\tilde{Z}_{a/c}$ are given by

$$
\tilde{Z}_{a,k} = Z_{a,k}([k^n + \tilde{k}^n]^{1/n}), \qquad \tilde{Z}_{c,k} = Z_{c,k}(k),
\tag{85}
$$

with $\tilde{k} = 1 \text{ GeV}$. The choice eq. (85) ensures that the regulators have the same (average) momentum scaling as the two-point functions, regulators proportional to the respective wave function renormalisations of the fields are RG-adapted, see [55]. Moreover, the scale $\tilde{k} = 1 \text{ GeV}$ is introduced for computations convenience; it leads to a gluon regulator, that does not diverge at $p = 0$ for $k \to 0$. While even a singular regulator choice at $p = 0$ does not contribute to the momentum integral, it complicates the numerics.

In [67] the regulator was used as it optimises fully momentum dependent approximations, see [55]. However, the resolution of eq. (82) requires the computation of $\operatorname{Tr} \mathcal{F}(-D^2)$ and $\operatorname{Tr} \mathcal{F}(-D_T)$ in terms of the discrete Eigenvalues or spectrum of the Laplacians $-D^2$ and $D_T$. The spectral properties of the Laplacians are discussed in Appendix C. see also [46].

The optimisation of the approximation in terms of its momentum dependence as used in [67] comes at the price that soft but sharp regulators delay the onset of the asymptotic ultraviolet scaling in the presence of a discrete momentum spectrum, see [103]. Here, asymptotic UV scaling entails, that the effective action reduces to the classical one with a running prefactor, see eq. (65). Indeed, for non-analytic regulators such as the Litim regulator or the sharp regulator the asymptotic UV scaling. In Appendix D we investigate the asymptotic UV scaling in the present set up as well as the regulator (in)dependence of our results.

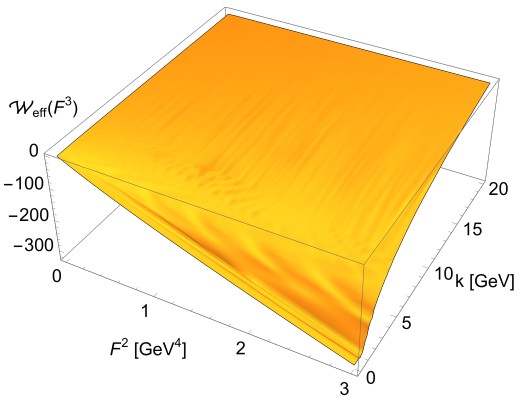

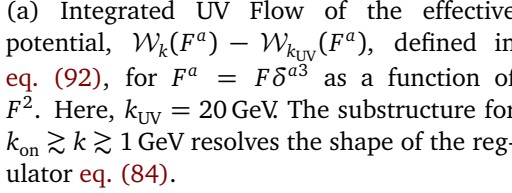

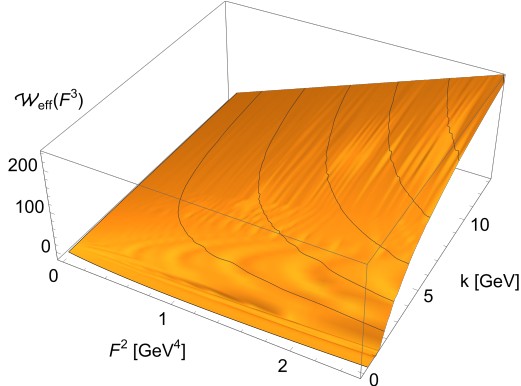

(a) Integrated UV Flow of the effective potential, $\mathcal{W}_k(F^a) - \mathcal{W}_{k_{\mathrm{UV}}}(F^a)$, defined in eq. (92), for $F^a = F\delta^{a3}$ as a function of $F^2$. Here, $k_{\mathrm{UV}} = 20\,\mathrm{GeV}$. The substructure for $k_{\mathrm{on}} \gtrsim k \gtrsim 1\,\mathrm{GeV}$ resolves the shape of the regulator eq. (84).

(b) Effective Potential $\mathcal{W}_k(F^a)$, defined in eq. (64), for $F^a = F\delta^{a3}$ as a function of $F^2$ in the regime $0 \le k \le k_{\mathrm{UV}} = 20\,\mathrm{GeV}$. The substructure for $k_{\mathrm{on}} \gtrsim k \gtrsim 1\,\mathrm{GeV}$ resolves the shape of the regulator eq. (84). For $k = 0$ see also Figure 1.

Figure 5: Cutoff dependence of the effective potential.

## C  Spectral properties of Laplacians

In this section we will comment on the background-covariant Laplacians, which were used for the momentum dependence of the Landau-gauge propagators in eq. (59) and eq. (82). Their explicit form follows from the gauge-invariant background field effective action [104] and is given by

$$\mathcal{D}_{T\,\mu\nu} = -D^2\delta_{\mu\nu} + 2ig\,F_{\mu\nu}, \quad \mathcal{D}_{L\,\mu\nu} = -D_\mu D_\nu, \tag{86}$$

and $\mathcal{D}_{\mathrm{gh}} = -D^2$. The transverse Laplacian also contains the spin-1 coupling to the background field.

The traces over the Laplace-type operators in eq. (82) can be evaluated upon introduction of Laplace transforms using standard heat-kernel techniques. The subtleties arising from the presence of a self-dual background are discussed in-depth in e.g. [68, 105, 106]. Here, we just quote the relevant spectra in self-dual backgrounds from [46],

$$\mathrm{Spec}\left\{-D^2\right\} = F_l(n + m + 1), \quad n, m = 0, 1, 2, \dots,$$

$$\mathrm{Spec}\left\{\mathcal{D}_{\mathrm{T}}\right\} = \begin{cases} F_l(n + m + 2) & , \quad \text{multiplicity} \quad 2 \\ F_l(n + m) & , \quad \text{multiplicity} \quad 2 \end{cases},$$

$$\tag{87}$$

where $F_l = |\nu_l|F/\sqrt{2}$. Here, dividing by $\sqrt{2}$ accounts for the multiplicity in a self-dual formulation of $F_{\mu\nu}$, and $\nu_l$ are the eigenvalues to the adjoint color matrix $n^a t^a$. The covariant spin-1 Laplacian $\mathcal{D}_{\mathrm{T}}$ has a double zero mode for $n = m = 0$ which is due to the symmetry between colour-electric and colour-magnetic field. The spectral problem of the longitudinal Laplacian $\mathcal{D}_{\mathrm{L}}$ can be mapped onto that of $-D^2$, such that eq. (87) is sufficient for the calculation in the main part of the paper, see e.g. [68, 105, 106]. The trace $\mathrm{Tr}'$ is defined as that without the zero

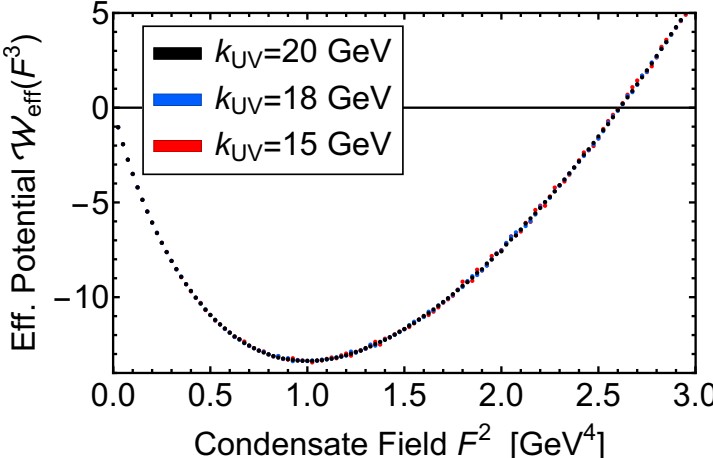

Figure 6: RG-consistency of the effective potential $\mathcal{W}_{\mathrm{eff}}(F^a)$: It is shown for integrating the initial effective potential eq. (89) at different initial cutoff-scales $k_{\mathrm{UV}} = 20, 18, 15\,\mathrm{GeV}$ to $k = 0$. The result is independent of the initial scale (RG-consistency).

mode, and for a general function $\mathcal{F}$ we get,

$$\mathrm{Tr}'\,\mathcal{F}(\mathcal{D}_{\mathrm{T}}) = 2\sum_{l=1}^{N_c^2-1}\left(\frac{F_l}{4\pi}\right)^2\left\{\sum_{n,m=0}^{\infty}\mathcal{F}\big(F_l(n+m+2)\big) + \sum_{n=0}^{\infty}\sum_{m=1}^{\infty}\mathcal{F}\big(F_l(n+m)\big) + \sum_{n=1}^{\infty}\mathcal{F}\big(nF_l\big)\right\}$$

$$= 4\sum_{l=1}^{N_c^2-1}\left(\frac{F_l}{4\pi}\right)^2\sum_{n,m=0}^{\infty}\mathcal{F}\big(F_l(n+m+1)\big) = 4\,\mathrm{Tr}_{xc}\,\mathcal{F}(-D^2), \tag{88}$$

where the trace Tr sums over momentum or space-time, internal indices and Lorentz indices of the respective field mode. Equation (88) displays an isospectrality relation between $-D^2$ and the non-zero eigenvalues of $\mathcal{D}_{\mathrm{T}}$. As a consequence, all gluon and ghost modes except for the two zero modes couple in the same fashion to the selfdual background. This allows us to compute eq. (82).

# D   UV Asymptotics of the effective potential and regulator independence

The present work utilises the ghost and gluon propagators from [47]; which has been obtained within a quantitative approximation to the full Yang-Mills system. There, and in respective works in QCD, [67,75,107] it has been checked that the choice of the regulator is of subleading importance for the propagators at vanishing cutoff scale, which is one of the self-consistency checks that goes into an estimate of the systematic error.

As mentioned at the end of Appendix B, the relatively sharp regulator here delays the onset of UV asymptotics and hence the onset cutoff scale $k \gtrsim k_{\mathrm{on}}$ of the regime in which the effective potential reduces to the classical form eq. (65). For the sake of convenience we recall it,

$$\mathcal{W}_k(F^a) \xrightarrow{k \gtrsim k_{\mathrm{on}}} \frac{F^2}{16\pi\alpha_s(k)}, \qquad \alpha_s(k) = \frac{1}{4\pi}\frac{g_s^2}{Z_{A,k}}, \tag{89}$$

with $Z_{A,k} = Z_{A,k}(p=0)$. In this regime the flow is simply a linear function in $F^2$ with the slope $\partial_t 1/(16\pi\alpha_s)$. Hence, for large cutoff scales we have,

$$\partial_t \mathcal{W}_k(F^a) \rightarrow -\frac{\partial_t \alpha_s(k)}{\alpha_s(k)} \frac{1}{16\pi\alpha_s(k)} F^2. \tag{90}$$

The coupling $\alpha_s$ in eq. (89) is the background coupling which has the same (two-loop) universal $\beta$-function as the fluctuation coupling $\alpha_{s,\text{fluc}} = g_s^2/(4\pi Z_a Z_c^2)$ computed in [47]. However, the equivalence of the perturbative $\beta$-functions still allows for a global rescaling $\alpha_s = \bar{\gamma}\,\alpha_{s,\text{fluc}}$ whose value is checked by comparing the two flows for $k \rightarrow k_{\text{UV}}$,

$$\bar{\gamma} = \lim_{F^2 \rightarrow 0} \frac{16\pi\alpha_{s,\text{fluc}}^2}{\partial_t \alpha_{s,\text{fluc}}} \frac{\partial_t \mathcal{W}_k}{F^2} \approx 1. \tag{91}$$

This fixes our initial condition, and in Figure 5 we show both, the respective integrated flow, Figure 5a, and the full cutoff dependent effective potential that also involves the initial condition, Figure 5b. The integrated flow from the UV scale $k_{\text{UV}} = 20\,\text{GeV}$ to a general cutoff scale $k$ is given by

$$\mathcal{W}_k(F^a) - \mathcal{W}_{k_{\text{UV}}}(F^a) = -\int_k^{k_{\text{UV}}} \frac{\mathrm{d}k}{k} \partial_t \mathcal{W}_k(F^a). \tag{92}$$

One clearly sees the linear dependence on $F^2$ for $k \rightarrow k_{\text{UV}}$. At lower scales $k \rightarrow k_{\text{on}}$ with $k_{\text{on}} \approx 14\,\text{GeV}$ the transition regime sets in, in which the integrated flow resolves the shape function. Finally, for physical cutoff scales $k \lesssim 1\,\text{GeV}$, the form of the shape function gets irrelevant and the integrated flow is getting smooth again. This shows very impressively that the information about the shape function is integrated out and disappears in the physical limit $k \rightarrow 0$.

We have also checked that the effective potential $\mathcal{W}_{\text{eff}}(F^a)$ is RG-consistent [55, 79]. This is the simple requirement that $\mathcal{W}_{\text{eff}}(F^a)$ does not vary if the flow is initiated at another cutoff scale $k_{\text{UV}}$. Accordingly, it is a consistency check on the initial effective potential $\mathcal{W}_{k_{\text{UV}}}$. Figure 6 depicts the physical effective potential $\mathcal{W}_{\text{eff}}(F^a)$, obtained from computations with $k_{\text{UV}} = 15, 18, 20\,\text{GeV}$. The initial effective potentials are given by eq. (89), where the scale dependency of the coupling $\alpha_s$ is obtained from the 1-loop beta function of the background coupling. These computations confirm the quantitative validity of the one-loop estimate for $\mathcal{W}_{k_{\text{UV}}}$ for these large initial cutoff scales. In turn, for lower cutoff scales, the one-loop form is gradually lost which can be easily seen by the substructure (in $F^2$) of the flow.

Finally, we also report on results for the effective potential obtained by integrating the flow with a smoother regulator

$$R_k(p) = k^2 e^{-p^2/k^2}. \tag{93}$$

Such a regulator decreases the numerical effort considerably. Note that this is not a self-consistent computation as it also requires cutoff-dependent propagators computed with the same regulator eq. (93). However, we use this as a stability test of our results, and hence a further systematic error control. The respective result for the cutoff dependent effective potential is shown in Figure 7, and one clearly sees that the use of a smoother regulator removes the substructures in the flow. The minimum value of $F^2$ at $k=0$ is given by

$$\langle F^2 \rangle_{\lambda_3} = 0.93(14)\,\text{GeV}^4, \tag{94}$$

to be compared with eq. (70). These values compare well, which informs our estimate of the systematic error.

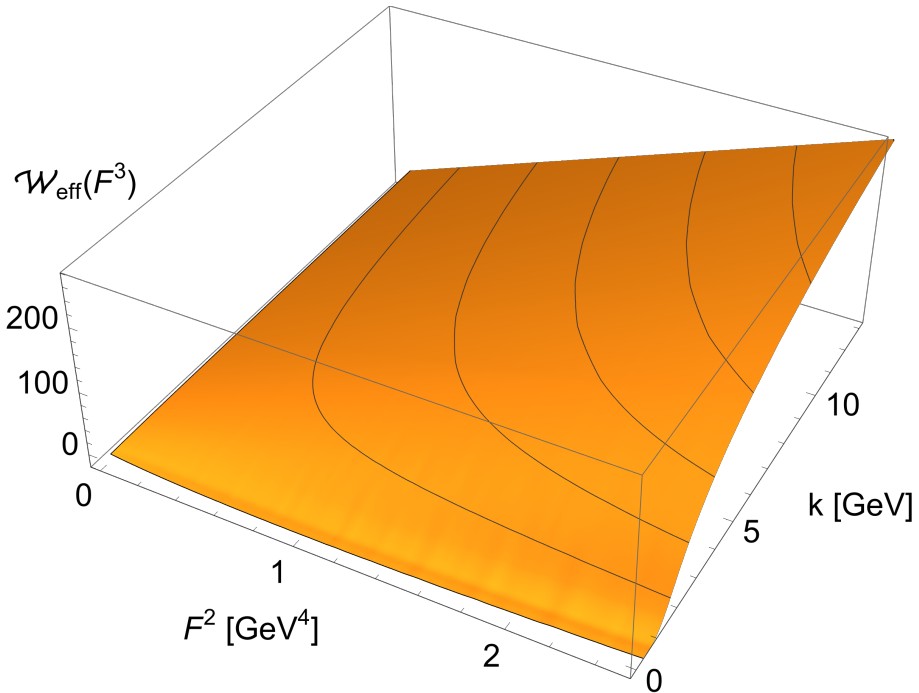

Figure 7: Effective Potential $\mathcal{W}_k(F^a)$, defined in eq. (64), for $F^a = F\delta^{a3}$ as a function of $F^2$ obtained from integrating the flow with the regulator eq. (93). In comparison to Figure 5b the regulator is much smoother, which translates to the smoothness in $k_{\text{on}} \gtrsim k \gtrsim 1\,\text{GeV}$.

## E   Fitting procedure

Formally, the coefficient $Z_{\text{cond}}$ in eq. (50) is defined via an operator product expansion of the gluon propagator, and stems from the local operator eq. (20). The present computation of the effective potential $\mathcal{W}_{\text{eff}}$ is detailed in Appendix D, Appendix C, Appendix B and uses the scaling propagator from [47]. The latter is obtained within a quantitative approximation of the coupled set of functional equations for Yang-Mills correlation functions, for respective DSE results see [72]. In [47], also decoupling solutions have been computed including a lattice-type solution, for respective lattice propagators see [20, 85].

The extraction of the $p^4$-coefficient stemming from eq. (20) requires the distinction of the infrared dynamics in the propagator, which in the present approach relates to the emergence

Table 1: Extrapolation results for the wave function renormalisation $Z_{\text{cond}}$ at $p = 0$ based on the fit results for $Z_{\text{cond}}(p_{\text{min}})$ as a function of the lower fit interval bound $p_{\text{min}}$, see Figure 8. The final estimate is obtained as the average of the scaling fRG and decoupling lattice data. In order to conservatively estimate possible systematic uncertainties (see text), we use the separate scaling fRG and lattice results as error bars.

|  | $Z_{\text{cond}}$ [GeV$^{-2}$] |
|---|---|
| scaling (fRG) | 0.168(31) |
| decoupling (lattice) | 0.129(19) |
| decoupling (fRG) | 0.1147(22) |
| Estimate | 0.149(19) |

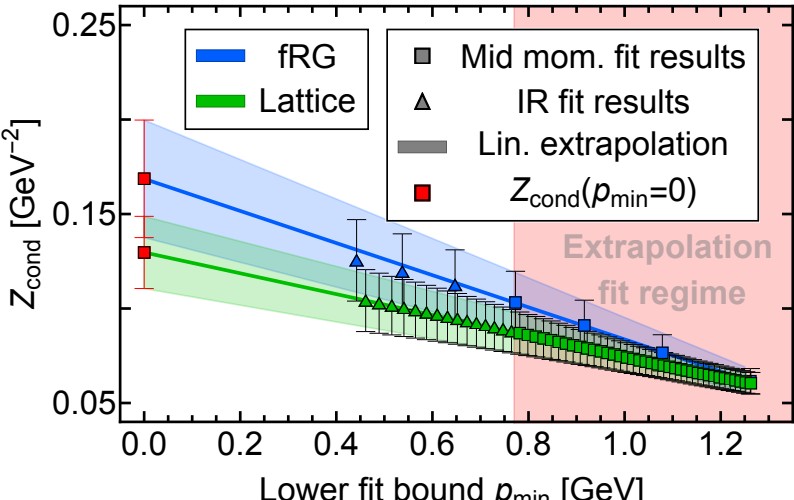

Figure 8: Linear extrapolation of $Z_{cond}$ to the lower fit interval bound $p_{min} = 0$, yielding $Z_{cond} = 0.149(19)$. The explicit fit results for $Z_{cond}$ are obtained via a fit of eq. (95) to the scaling fRG data of [47] (blue squares), and to the lattice data of [22, 85] (green squares). $Z_{cond}$, being defined as the operator product expansion coefficient should be extracted at $p = 0$: we extract this information from an extrapolation of the fit results towards $p = 0$ (red squares), and use as a minimal $p_{min} \approx 0.8$ GeV, below which the details of the implementation of the IR dynamics begin to matter. The triangular data points mark fit results for $p_{min}$ below the fit regime for the interpolation. The final estimate for $Z_{cond}$ (eq. (99) and eq. (74)) is obtained as the mean of the lattice and scaling fRG results for $Z_{cond}$, whose numerical values can be found in Table 1.

of the color condensates, from the coefficients of the local operators. This mixing for small momenta makes it impossible to extract the $p^4$-coefficient in an expansion about $p = 0$ without further information on the momentum dependence of the condensate. Instead we shall evaluate the propagator for sufficiently large momentum scales, for which the condensate vanishes, $\langle F \rangle \to 0$. The cutoff scale resembles the momentum scale $p$, indeed it is introduced in the two-point function itself as a momentum cutoff. Hence, we deduce from the flow of the minimum of $\mathcal{W}_{eff}$ depicted in Figure 3, that the condensate vanishes for $p \gtrsim 1/2$ GeV. Accordingly we determine $Z_{cond}$ from fits

$$Z_a^{fit}(p^2) = \frac{Z_m}{p^2} + Z_{p^2} + Z_{cond}\, p^2 \tag{95}$$

to the gluon wave function $Z_a(p^2)$ in the momentum regime

$$p \in [p_{min}, p_{max}], \tag{96}$$

with

$$p_{min} \in [0.77, 1.27]\, GeV, \quad p_{max} \in [1.95, 2.23]\, GeV, \tag{97}$$

where the range of values for $p_{min}$ is adapted to the data points of the sparse fRG data.

The upper bound $p_{max}$ is chosen such, that the interval sustains a Taylor expansion while containing a sufficient amount of data points for fitting, also adapted to the fRG data points. Its maximum value is further constrained by the UV boundary of the lattice data from [85], which are used for comparison as well as the error estimate, together with the lattice data from [22].

The constants $Z_m$, $Z_{p^2}$ and $Z_{\text{cond}}$ in eq. (95) are fit parameters. Here $Z_m$ takes care of the infrared gapping dynamics, and $Z_{p^2}$ related to a standard (infrared) wave function renormalisation. Both parts carry the details of the IR behaviour of the propagator and may vary largely for different solutions. In turn, the coefficient $Z_{\text{cond}}$ should not.

We perform the fits for different values of the lower fitting interval bound $p_{\text{min}}$. For every fit, $p_{\text{max}}$ is varied between the points in the $p_{\text{max}}$ interval, comp. eq. (97). In addition, we transform the lattice and fRG data sets into the respective (inverse) dressing function and inverse propagator, and fit those with the respective fit functions corresponding to eq. (95). This provides us with a $Z_{\text{cond}}(p_{\text{min}})$ given as the average over the single fit results for the different values of $p_{\text{max}}$ and representations of the data set, with uncertainty given by the standard deviation.

Eventually, we extract the wave function renormalisation $Z_{\text{cond}}$ at $p = 0$ via a limiting procedure as

$$Z_{\text{cond}} = \lim_{p_{\text{min}} \to 0} Z_{\text{cond}}(p_{\text{min}}). \tag{98}$$

The limit is obtained within an extrapolation of the $Z_{\text{cond}}(p_{\text{min}})$ discussed below. We extract $Z_{\text{cond}}$ from both the scaling fRG data of [47] as well as the lattice solution [85], see Figure 8 and Table 1 for the numerical values. We also provide $Z_{\text{cond}}$ from a lattice-type fRG decoupling solution for comparison in Table 1. When lowering the lower fit interval bound $p_{\text{min}}$, the results for $Z_{\text{cond}}$ differ more and more. This can be attributed to the different infrared behaviour of the two data sets. Accordingly, we exclude as many incompatible data points as possible from the extrapolation fit regime while keeping enough data for a meaningful prediction of $Z_{\text{cond}}(p = 0)$.

As the data from [47] are relatively sparse and hence the respective $Z_{\text{cond}}(p_{\text{min}})$ and the extrapolation show large error bars, we support this extrapolation with one obtained from dense fRG data provided in [108, 109]. While the approximation used in the latter computations is not as sophisticated as that used in [47], it allows for a relatively quick production of dense data. The scaling solution of [108] yields $Z_{\text{cond}} = 0.166(33)$, which agrees extremely well with the scaling solution estimate of [47], comp. Table 1.

Our final estimate for $Z_{\text{cond}}$ is obtained by averaging the scaling fRG and lattice result, yielding

$$Z_{\text{cond}} = 0.149(19). \tag{99}$$

The error bars are given by the separate extrapolation results for scaling fRG and lattice data in order to incorporate systematic uncertainties such as the influence of the different infrared behaviours.

# F   Schwinger mechanism

In order to facilitate the comparison with the literature, in this Appendix we modify the notation employed in the main body of the article, denoting by $\Delta(q^2)$ and $D(q^2)$ the gluon and ghost propagators, respectively, and by $\mathcal{Z}(q^2)$ and $F(q^2)$ their dressing functions: $\mathcal{Z}(q^2) := q^2 \Delta(q^2)$ and $F(q^2) := q^2 D(q^2)$.

According to one of the main approaches put forth in a number of works [99, 110–113], the generation of an effective gluon mass proceeds through the non-Abelian implementation of the well-known Schwinger mechanism [114–117]. Within this scenario, the fundamental vertices that enter in the DSE of the gluon propagator, $\Delta(q^2)$, contain longitudinally coupled massless poles, which eventually trigger the result $\Delta^{-1}(0) := m_{\text{gap}}^2$.

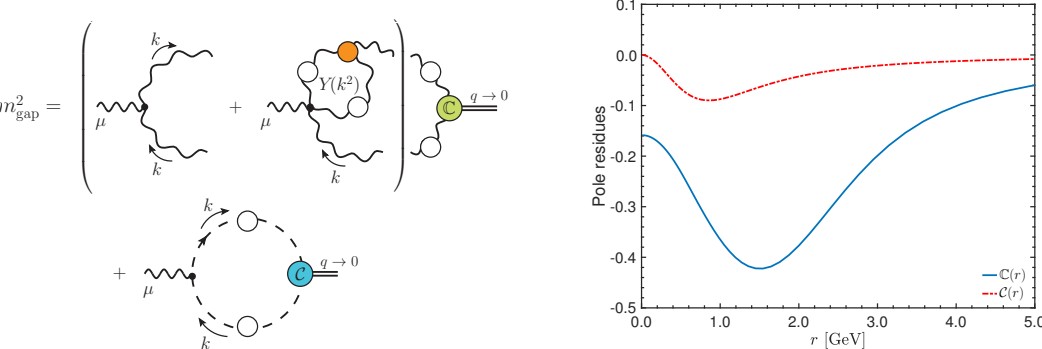

Figure 9: Left panel: Diagrammatic representation of eq. (103). Right panel: the momentum dependence of $\mathbb{C}(q^2)$ and $\mathcal{C}(q^2)$.

In particular, the three-gluon vertex, $\mathbb{\Gamma}_{\mu\alpha\beta}(q,r,p)$, and the ghost-gluon vertex, $\mathbb{\Gamma}_{\mu}(q,r,p)$, are composed by two distinct types of terms, namely

$$\mathbb{\Gamma}_{\mu\alpha\beta}(q,r,p) = \Gamma_{\mu\alpha\beta}(q,r,p) + \frac{q_\mu}{q^2} g_{\alpha\beta} C_1(q,r,p) + \cdots,$$

$$\mathbb{\Gamma}_{\mu}(q,r,p) = \Gamma_{\mu}(r,p,q) + \frac{q_\mu}{q^2} C(q,r,p), \tag{100}$$

where the terms $\Gamma_{\mu\alpha\beta}(q,r,p)$ and $\Gamma_{\mu}(q,r,p)$ contain all pole-free contributions, which may diverge at most logarithmically as $q \to 0$ [118]. The ellipses in the first relation of eq. (100) denote terms proportional to $r_\alpha/r^2$ or $p_\beta/p^2$, which are annihilated when contracted with the transverse (Landau gauge) gluon propagators inside the relevant diagrams of the DSEs, or tensorial structures that are subleading in the limit $q \to 0$.

A detailed analysis [119] based on the Slavnov-Taylor identities satisfied by the above vertices reveals that

$$C_1(0,r,-r) = C(0,r,-r) = 0. \tag{101}$$

Therefore, the Taylor expansion of $C_1(q,r,p)$ and $C(q,r,p)$ around $q = 0$ yields

$$\lim_{q\to 0} C_1(q,r,p) = 2(q\cdot r)\underbrace{\left[\frac{\partial C_1(q,r,p)}{\partial p^2}\right]_{q=0}}_{\mathbb{C}(r^2)} + \mathcal{O}(q^2), \tag{102}$$

$$\lim_{q\to 0} C(q,r,p) = 2(q\cdot r)\underbrace{\left[\frac{\partial C(q,r,p)}{\partial p^2}\right]_{q=0}}_{\mathcal{C}(r^2)} + \mathcal{O}(q^2).$$

Thus, inserting the vertices of eq. (100) into the DSE of the gluon propagator and taking the limit $q \to 0$, one arrives at (see Appendix E) [112]

$$m_{\text{gap}}^2 = \frac{3C_A\alpha_s}{8\pi}\int_0^\infty dy\, \mathcal{Z}^2(y)[6\pi\alpha_s C_A Y(y) - 1]\mathbb{C}(y) + \frac{C_A\alpha_s}{8\pi}\int_0^\infty dy\, F^2(y)\mathcal{C}(y). \tag{103}$$

In the above formula, $\alpha_s = g_s^2/4\pi$, defined at the renormalisation point $\mu$ where the ingredients of eq. (103) have been renormalised, within the momentum subtraction (MOM) scheme; the renormalisation point has been chosen at $\mu = 4.3$ GeV. Moreover, $C_A$ is the Casimir eigenvalue of the adjoint representation with $C_A = N_c$ for $SU(N)$. Finally, $\mathcal{Z}(y)$ and $F(y)$ denote the

dressing functions of the gluon and ghost, respectively, and $Y(k^2)$ is an appropriately projected contribution of the subdiagram shown in Appendix E.

The functional form of the pole residues $\mathbb{C}(k^2)$ and $\mathcal{C}(k^2)$ is determined from the linear homogeneous system of coupled Bethe-Salpeter equations that they satisfy. This system is derived from the corresponding DSEs governing the dynamics of $\mathbb{II}_{\mu\alpha\beta}(q,r,p)$ and $\mathbb{II}_{\mu}(q,r,p)$, in the limit $q \to 0$; for further details, see [112].

The resulting eigenvalue problem yields non-trivial solutions for $\mathbb{C}(k^2)$ and $\mathcal{C}(k^2)$, for a specific value of the coupling $\alpha_s$, which depends on the details of the ingredients that enter in the kernels of the Bethe-Salpeter system. It is important to emphasise that the homogeneity and linearity of the equations leaves the overall scale of the corresponding solutions undetermined. The scale setting is implemented by solving the vertex DSEs for *general kinematics*, using as input the particular $\alpha_s$ that was singled out by the eigenvalue condition. Then, from the general 3-D solution the particular slice that corresponds to $\mathbb{C}(k^2)$ and $\mathcal{C}(k^2)$ is identified, furnishing precisely the correctly rescaled version of the solutions obtained from the system. The final form of the scale-fixed pole residues is shown in Appendix E.

The next step consists in substituting into eq. (103) the scale-fixed $\mathbb{C}(k^2)$ and $\mathcal{C}(k^2)$, and use refined lattice data [85] for the gluon and ghost dressing functions, $\mathcal{Z}(k^2)$ and $F(k^2)$. The lattice propagators have been normalised at the point $\mu = 4.3$ GeV, namely the highest momentum scale available in this simulation. For the purpose of the comparison with the results computed in the present work we match the scales of the lattice data in [85] with that in [47], which leads us to

$$m_{\text{gap}}^{(\text{Schwinger})} = 0.320(35)\,\text{GeV}. \tag{104}$$

Equation (104) is in excellent agreement with the estimate $m_{\text{gap}} = 0.322(34)\,\text{GeV}$ obtained in the present work, see eq. (75). Both compare rather favourably to the central lattice value $\Delta^{-1/2}(0) = 0.354$ GeV. The predominant source of error in the calculation using the Schwinger mechanism originates from the uncertainties in the non-perturbative structure of the pole-free vertex $\Gamma_{\mu\alpha\beta}(q,r,p)$, which affects both the determination of the function $Y(k^2)$ in eq. (103), as well as the kernels of the Bethe-Salpeter equations that determine the functions $\mathbb{C}(k^2)$ and $\mathcal{C}(k^2)$.

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
