# Peer review of "Gluon condensates and effective gluon mass"

_SciPost Physics, doi:SciPost Phys. 13, 042 (2022)_

## Round 1 · Referee Report · Anonymous (Referee 2) · 2022-3-28

Report

Here we present a referee report for the paper ``Gluon condensates and effective gluon mass'' by Jan Horak, Friederike Ihssen, Joannis Papavassiliou, Jan M. Pawlowski, Axel Weber, and Christof Wetterich.

In this present manuscript, the authors compute the effective potential for a constant field strength $F_{\mu\nu}$, whose non-trivial minimum is associated with the formation of a color condensate. In the sequence, they establish a connection between the formation of the color gluon condensate to the emergence of an effective gluon mass in the Yang-Mills Green's functions, performing an average over color directions. As a result, the authors found that the condensate value $\langle F \rangle^2$ is in good agreement with previous phenomenological estimates. In addition, the determination of the scale of the gluon mass agrees rather well with previous the lattice and Schwinger-Dyson results.

As stressed throughout the manuscript, the present analysis may be faced as a starting point for a systematic exploration of the connection between gluon condensates and the gluon mass gap. The two crucial simplified working hypotheses were: {\it(i)} compute the gauge-invariant effective potential for constant field strength $F_{\mu\nu}$ and {\it(ii)} the color-averaging procedure, which leads to the appearance of averaging factor $f_{av}(N_c)$ in the gluon mass expression.

In my opinion, the general idea and the manuscript results are quite interesting. For these reasons, the article merits publication in SciPost. However, before that, the authors should clarify the following point

One of the authors' main results is the relation between the gluon mass and the condensate $\langle F \rangle^2 $ given by Eq.~(34). Back in 88, using operator product expansion, Lavelle & Schaden in Ref. [41] found a similar color factor connecting the gluon self-energy and the condensate $\langle G \rangle^2 $. Since the color averaging procedure is admittedly the largest source of systematic error in the present work, it would be nice if the authors included a paragraph saying that a similar color structure was found previously in [41] and discussed if there is any reason for such different methods lead to precisely the same color factor.
  • validity: high
  • significance: high
  • originality: high
  • clarity: high
  • formatting: good
  • grammar: excellent

Author:  Jan Horak  on 2022-04-06  [id 2362]

(in reply to Report 2 on 2022-03-28)

The authors thank the referee for his report.

The referee adresses that color averaging factor in eq. (34) of the manuscript is identical to that one found by Lavelle & Schaden in [41]. In the expression for the gauge-invariant effective potential, eq. (18) of [41], the factor $(N_c^2 - 1)/N_c$ indeed appears, but is multiplying the quark condensate $\langle \bar \psi \psi \rangle$, which we do not consider. The color prefactor in front of the gluon condensate $\langle G^2 \rangle$ (where G is the field strength) is $N_c$ however, which is not identical to our result. In fact, [41] finds the same linear large $N_c$-scaling of the condensate term in the effective potential as we do particular, which we found to be an important consistency check for our color averaging procedure, cf. Sec. IIIc.

The inverse of our color factor $(N_c^2 - 1)/N_c$ also appears in front of the $\langle G^2 \rangle$ condensate in the gluon self-energy in eq. (11) of [41]. This we consider to be coincidental.

---

## Round 1 · Referee Report · Anonymous (Referee 1) · 2022-3-28

Report

The authors investigate a scenario for the origin of the infrared behavior of the gluon propagator in covariant gauges in the form of an effective mass-like behavior as observed in many nonperturbative studies both on the lattice as well as in the continuum. They propose a Higgs-like mechanism generated dynamically by the theory where color condensates play the role of a Higgs field. Their study provides evidence for this scenario on the basis of a nonperturbative estimate of the effective action using the functional RG. Remarkably, the quantitative result for their estimate of the gluon-mass parameter obtained from their continuum method compares favorably to lattice results and other approaches.

This paper successfully combines two key ideas: the phenomenological description of low-energy properties of QCD using color condensates on the one hand, and the possibility to compute phenomenologically relevant quantities based on nonperturbative propagators and the functional RG in background formalism on the other. In this way, the so far existing gap of a conceptual understanding for the gluon mass behavior observed in Landau-gauge propagators has been closed; or at least, a promising way for closing this gap has been identified.
The paper is very well written, technical details are comprehensively explained in the Appendices and the manuscript has been thoroughly prepared. I can fully recommend the paper for publication in SciPost.

I have one question which the authors may wish to address in the final version:

In the par. below (70), the authors claim that the result of eq.(70) should be considered as an upper bound, since it is "composed by the condensates of both $F^2$ and $F\tilde{F}$".
This statement seems confusing: I understand that the self-dual background field does not allow a clean distinction between operators built from $F^2$ and those built from $F\tilde{F}$. Therefore, an effective action (density) of the form W($F^2$,$F\tilde{F}$) is (mis-)identified as $W$($F^2$,$F\tilde{F}$=$F^2$) $= W$($F^2$). This makes clear that the desired effective potential for $F^2$ may be contaminated by terms arising from $F\tilde{F}$ operators. However, I do not see that this necessarily leads to an overestimation of the condensate. Corrections could equally go into the opposite direction.

Also, I am not certain whether the authors indeed wanted to make the statement that they expect the occurrence of an $F\tilde{F}$ condensate. Wouldn't such a condensate be an unwanted source of CP violation? (Of course, a local topological charge density is expected to be nonzero, but I would assume that it averages to zero globally in order to preserve CP.)

Requested changes

I also found a couple of typos which the authors may wish to correct:

  • Eq.(19): "$...+ a_\mu$" --> "$...+ a_\mu^a$"
  • line below Eq.(19): "... as the fields strength." --> "... as the field strength."
  • Par. below Eq.(27): "At this state ..." --> "At this stage ..."
  • 1st par., right col. on p.5: "... color blind strength..." --> "... color blind value ..." ?
  • below (39c): "... function of the Laplacian $D$ ..." --> "... function of the covariant derivative D ..."
  • 1st sentence, right col on p.12: "Now we us ..." --> "Now we use ..."
  • Ref. [107] is incomplete.

  • validity: -
  • significance: -
  • originality: -
  • clarity: -
  • formatting: -
  • grammar: -

Author:  Jan Horak  on 2022-04-06  [id 2361]

(in reply to Report 1 on 2022-03-28)

The authors thank the referee for his report and careful reading of our manuscript.

The referee adresses that the possible contribution of $F \tilde F$-terms to our condensate value eq. (70) might as well be negative. Furthermore, such a contribution represents a source of CP violation, as the referee points out.

We softened the statement below eq. (70) about the direction $F \tilde F$-contribution to our result and added a comment about the CP-violating nature of such contributions. The typos listed under requested changes have been fixed.

---

## Round 2 · Referee Report · Anonymous (Referee 2) · 2022-4-8

Report

The authors have addressed properly my queries in the updated version of the manuscript. For this reason, I recommend publication.

---

## Round 2 · Referee Report · Anonymous (Referee 1) · 2022-4-9

Report

Since the authors have fully addressed my minor points in the updated version, I am delighted to recommend publication.

---

## Round 2 · List of Changes

• removed comment about direction of F F dual contribution
  • corrected typos as requested

---

## Editorial Decision

published